# A magnetic multi-layer soft robot for on-demand targeted adhesion

Ziheng Chen [1,2,3,8], Yibin Wang[2,3,8], Hui Chen[2,3], Junhui Law [2,4], Huayan Pu [1], Shaorong Xie [5], Feng Duan[6], Yu Sun [4], Na Liu [1] ✉ & Jiangfan Yu [2,3,7] ✉

Magnetic soft robots have shown great potential for biomedical applications due to their high shape reconfigurability, motion agility, and multi-functionality in physiological environments. Magnetic soft robots with multi-layer structures can enhance the loading capacity and function complexity for targeted delivery. However, the interactions between soft entities have yet to be fully investigated, and thus the assembly of magnetic soft robots with on-demand motion modes from multiple film-like layers is still challenging. Herein, we model and tailor the magnetic interaction between soft film-like layers with distinct in-plane structures, and then realize multi-layer soft robots that are capable of performing agile motions and targeted adhesion. Each layer of the robot consists of a soft magnetic substrate and an adhesive film. The mechanical properties and adhesion performance of the adhesive films are systematically characterized. The robot is capable of performing two loco-motion modes, i.e., translational motion and tumbling motion, and also the on-demand separation with one side layer adhered to tissues. Simulation results are presented, which have a good qualitative agreement with the experimental results. The feasibility of using the robot to perform multi-target adhesion in a stomach is validated in both ex-vivo and in-vivo experiments.

Magnetic soft robots are promising candidates for biomedical applications[1-6]. They can be magnetically programmed with required profiles, and have high degrees of freedom in shape deformation[7-11], which is necessary for dexterous motions, such as grasping[9] and crawling[12]. By applying external magnetic fields, the soft robots with tailored magnetization profiles can traverse complex terrains by altering locomotion modes, such as rolling[12,13], crawling[12,14,15], and swimming[12,16]. By designing an actuation method and optimizing energy bursting, the jumping motion of magnetic soft robots has also been achieved in unstructured aquatic-terrestrial environments[17]. Moreover, locomotion can be realized in unstructured three-dimensional environments by considering surface microstructures

and surface coating of the robots, such as microspikes[18,19] and mucoadhesive film[20] loaded by magnetic soft robots.

Because of their high motion controllability and shape reconfigurability, magnetic soft robots have been used to perform biomedical tasks like minimally-invasive operation[21-24] and targeted delivery[25-29]. Bioprinting with a minimally invasive manner on a rat liver in vivo has been demonstrated[21], and the assistance of urination via applying mechanical compression to the underactive bladders has been achieved[22]. Meanwhile, magnetic soft robots have been utilized for cargo delivery in different organs and cavities, such as gastrointestinal (GI) tract[15,30-33] and blood vessel[26,34]. Releasing drugs by compressing capsule-shaped magnetic soft robots has been reported in an ex-vivo

[1]School of Mechatronics Engineering and Automation, Shanghai University, Shanghai 200444, China. [2]School of Science and Engineering, The Chinese University of Hong Kong, Shenzhen 518172, China. [3]Shenzhen Institute of Artificial Intelligence and Robotics for Society, Shenzhen 518172, China. [4]Department of Mechanical and Industrial Engineering, University of Toronto, Toronto, ON M5S 3G8, Canada. [5]School of Computer Engineering and Science, Shanghai University, Shanghai 200444, China. [6]Department of Interventional Radiology, Chinese PLA General Hospital, Beijing 100853, China. [7]School of Medicine, The Chinese University of Hong Kong, Shenzhen 518172, China. [8]These authors contributed equally: Ziheng Chen, Yibin Wang. ✉e-mail: liuna_sia@shu.edu.cn; yujiangfan@cuhk.edu.cn

pig stomach[30,35]. A magnetic modular soft robot enables the successful delivery of a therapeutic patch onto an ex-vivo porcine stomach ulcer[33]. Furthermore, a wireless magnetic soft stent incorporating multiple drug-loading structures has been proposed to perform on-demand and local release of drugs in phantoms[34].

Gastric ulcer is a common disease occurring in different locations in the GI tract, such as the gastric angle, gastric antrum, and cardia[36]. Multiple gastric ulcers can occur at the same period[37], and oral-taking drugs are widely used to treat ulcers. The therapeutic efficacy of oral-taking drugs is limited due to their low chemical stability in gastric fluid and shallow penetration in mucosa[38]. Bioadhesive platforms, such as patches[39–41] and hydrogels[42–44], have been reported to prolong drug retention by improving tissue adhesion in the porcine stomach. Magnetic soft robots offer an efficient and noninvasive approach for delivering bioadhesive platforms to gastric ulcers, which can mitigate erosion and enhance the healing effect by covering gastric ulcers[42,43]. Furthermore, using a magnetic soft robot with multi-layer structures has the potential in on-demand adhesion at different ulcer sites through the separation between layers of the robot. However, to date, the interactions between layers of the robot have yet to be fully investigated, challenging the realization of a magnetic multi-layer soft robot with on-demand motion modes.

Herein, we design a magnetic multi-layer soft robot by tailoring the magnetic interaction between layers, which is capable of performing navigated locomotion on biological tissues and multiple on-demand targeted adhesion at different sites. The robot consists of three layers, and each layer comprises a soft magnetic substrate with the magnetization direction perpendicular to its surface and an adhesive film capable of forming adhesion to wet tissues through hydrogen bonds. The adhesion performance of the adhesive film on ex-vivo porcine gastric tissues are characterized. Two locomotion modes of the robot on gastric tissues are explored and we demonstrate the on-demand separation between the robot and the adhered side layer through applying sufficient magnetic torque. The feasibility of using a magnetic multi-layer soft robot for on-demand targeted adhesion in an unstructured environment is validated in both ex-vivo gastric tissue and the stomach filled with fluid. The in-vivo trials of multi-target adhesion are also performed to validate the proposed strategy. This work presents an effective design of magnetic multi-layer soft robots, paving the way for promising clinical opportunities, specially for the operations in GI tract.

## Results

### Design of a magnetic multi-layer soft robot

The magnetic multi-layer soft robot has three layers, i.e., one center layer and two side layers, and each layer consists of a soft magnetic substrate and an adhesive film capable of forming adhesion with wet tissues (Fig. 1a). Each layer is magnetized with the same direction, i.e., perpendicular to its surface, for stable assembly through interlayer magnetic attraction. To achieve the adhesion between the side layer and the tissue, and also the layer-layer separation of the robot, soft magnetic substrates with distinct in-plane structures are designed. The soft magnetic substrate of the side layer has a magnetic frame and a nonmagnetic base, while the center layer has a nonmagnetic frame and a magnetic base. The magnetic base of the center layer can exert magnetic attraction forces to the side layers, in order to maintain the integrity of the robot during locomotion. Meanwhile, to generate magnetic pressing forces between the side layer and the tissue when

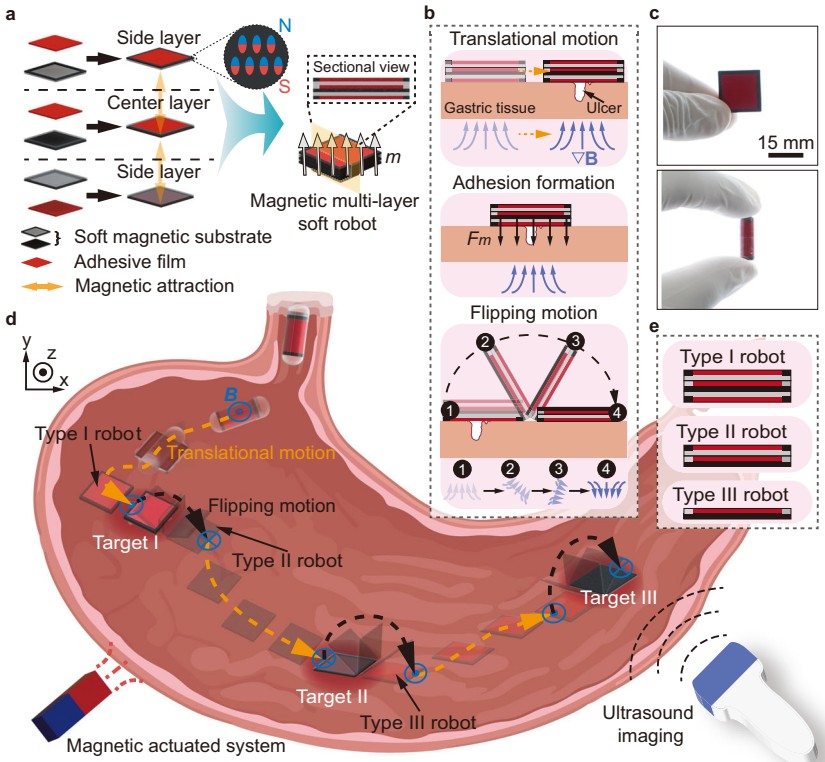

**Fig. 1 | Schematics of a magnetic multi-layer soft robot and its application for on-demand multi-targeted adhesion. a** Schematics of a magnetic multi-layer soft robot. The soft magnetic substrate of the side layer has a magnetic frame and a nonmagnetic base, while that of the center layer has a nonmagnetic frame and a magnetic base. **b** Schematic process of on-demand targeted adhesion to gastric ulcer. **c** A robot in released state and encapsulated state. **d** Overview of multi-target adhesion using the robot in a stomach under the tracking of ultrasound imaging. The orange dashed arrows represent the translational motion direction of the robot. The black dashed arrows represent the flipping motion direction of the robot. **e** Schematic expression of the structures of Type I, Type II and Type III robot. Parts of (**d**) are created with BioRender.com.

the robot reaches the target to accelerate the formation of adhesion is also a purpose. Furthermore, the magnetic interaction induced between the magnetic parts of the side layer and the center layer, i.e., the magnetic frame and the magnetic base, is weaker than the adhesive interaction formed between the side layer and the tissue, which facilitates the separation between layers of the robot. Besides the above design principles, an adhesion and separation strategy is proposed, as shown in Fig. 1b. The robot approaches the targeted ulcer on a gastric tissue through performing translational motion, and the adhesive film of the side layer then adheres to the gastric tissue by applying magnetic gradient force $F_m$. By altering the direction of the magnetic field, the magnetic torque exerted on the center layer overcomes the interlayer attraction, causing the robot to flip over while the adhesion between the side layer and the tissues is still maintained. In this work, the soft magnetic substrate comprises ferromagnetic particles NdFeB and polydimethylsiloxane (PDMS), while the adhesive film consists of Carbopol, which serves as mucoadhesive material[45,46]. Poloxamer and Hydroxypropylmethylcellulose (HPMC) are also added to the mucoadhesive material, contributing to the formation of films. Fabrication details of the robot are explained in Supplementary Fig. 1 and Methods.

The robot can be curled and encapsulated (Fig. 1c), facilitating its ingestion for accessing the gastrointestinal (GI) tract. When the robot enters the stomach, it can be navigated to targets by ultrasound imaging feedback and achieve on-demand targeted adhesion by implementing the adhesion and separation strategy. The process of multi-target adhesion, from target I to target III, is schematically illustrated in Fig. 1d. The robot achieves targeted adhesion at each stage, and the layers forming the robot are gradually reduced, ultimately leaving a free center layer for the adhesion to target III. We hereby term the three stages of the multi-layer robots as the Type I robot with a three-layer structure, the Type II robot with a two-layer structure, and the Type III robot with a single-layer structure (Fig. 1e).

## Mechanical properties of the adhesive film

The release process of the robot in the simulated gastric fluid (SGF) from a capsule is demonstrated in Fig. 2a. Schematics of the adhesion mechanism of the adhesive film is shown in Fig. 2b. The side layer of the robot contacts and adheres to the tissue using an adhesive film. The intermolecular hydrogen bonds between the three components (Carbopol, Poloxamer, and HPMC) facilitate the formation of an adhesive film. Carbopol, a polyacrylic acid polymer commonly used for mucoadhesive, plays a crucial role in the adhesive properties. The formation of hydrogen bonds between the carboxylic acid functional group of Carbopol and the glycoprotein component of the mucosa significantly contributes to the adhesion process[47]. The adhesive film can form adhesion to wet gastric tissue through physical cross-linking hydrogen bonds under pressure.

The mechanical properties of the adhesive film are measured by tensile and amplitude scanning tests (Methods). Five groups of adhesive films with different weight ratios ($\gamma$), from 1 : 6 to 5 : 6, of Carbopol to HPMC-Poloxamer mixture (i.e., the mixture of HPMC and Poloxamer) are prepared. The experimental results of tensile tests reveal that, the adhesive film has the maximum fracture stress and strain with a $\gamma$ of 2: 3, i.e., $23.08 \pm 2.77$ MPa and $21.04 \pm 4.44\%$, respectively (Fig. 2c). The results of amplitude scanning tests are presented in Fig. 2d, to show the change of the storage modulus (G′) and loss modulus (G″) of the adhesive film with applied strain (from 0.01% to 100% strain). Within the experimental measurements, the G′ is higher than the G″ when $\gamma$ is kept the same, indicating the adhesive films have not undergone shear fracture[48]. Among the five groups, the adhesive film fabricated with a $\gamma$ of 2: 3 exhibited the highest G′ and G″.

## Adhesion performance

To quantitatively evaluate the adhesion performance of the adhesive film, Lap-shear test and T-peel test are conducted to characterize the shear strength and interfacial toughness (Fig. 3a and Methods). The

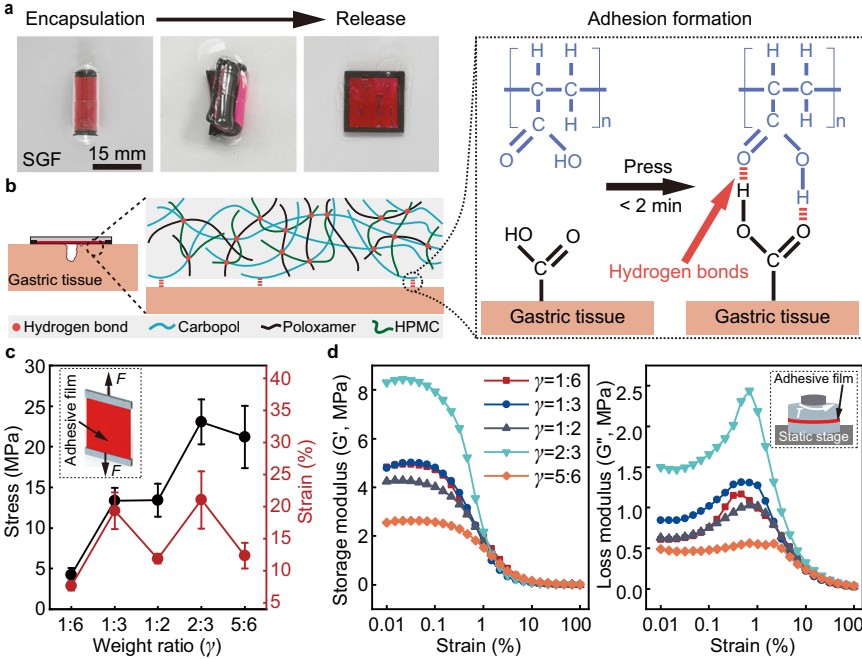

**Fig. 2 | Design and characterization of the adhesive film. a** The releasing process of a magnetic multi-layer soft robot from the encapsulated state. **b** Adhesion mechanism of the adhesive film. The adhesive film is mainly crosslinked by intermolecular hydrogen bonds between Carbopol, Poloxamer, and HPMC. The adhesion between the adhesive film and gastric tissue surface is formed through hydrogen bonds. **c** Tensile stress–strain curves of the adhesive film prepared with five different weight ratios of Carbopol and HPMC-Poloxamer mixture. The error bars are obtained from 3 trials in each condition. **d** The change of storage modulus and loss modulus of the adhesive films with strain. The films are prepared with five different weight ratios between Carbopol and HPMC-Poloxamer mixture. In all adhesive films, the HPMC-Poloxamer mixture is prepared with a weight ratio of 1 : 1 (HPMC and Poloxamer). Data are presented as mean values ± SD. Source data are provided as a Source Data file.

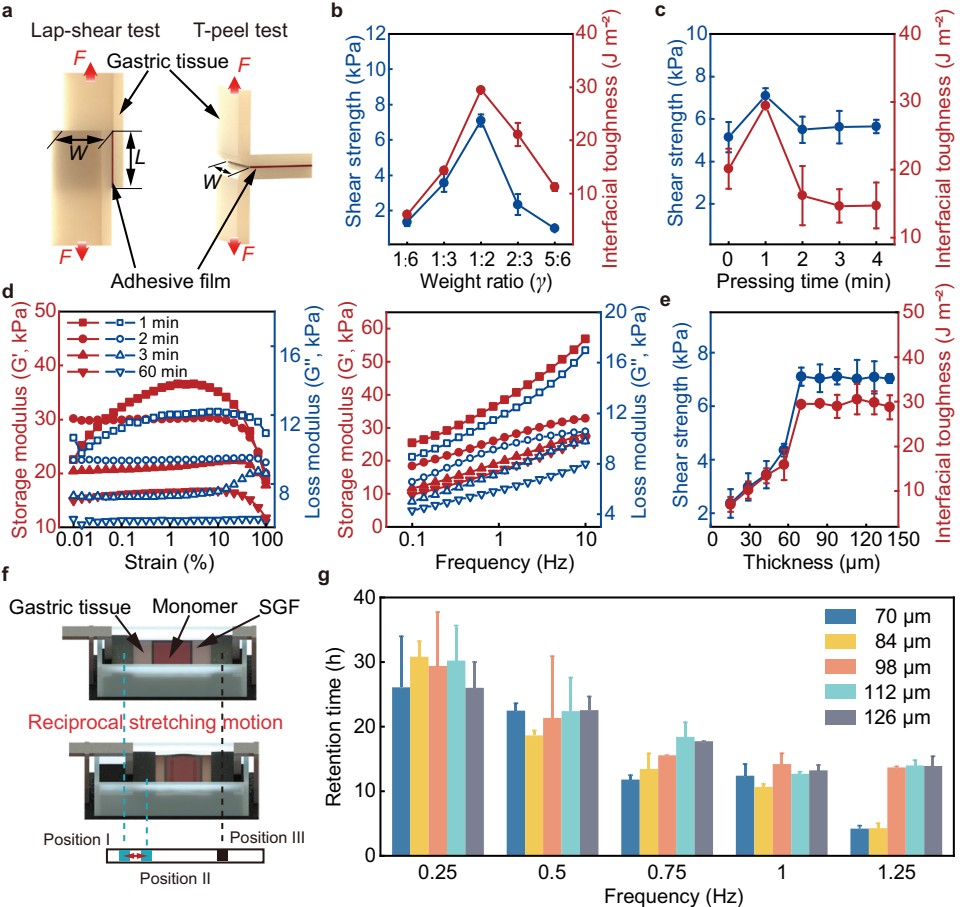

**Fig. 3 | Characterization of the adhesion performance. a** Schematics of the Lap-shear test and T-peel test. **b** Shear strength and interfacial toughness of the adhesive films prepared with five different ratios of Carbopol and HPMC-Poloxamer mixture. **c** Shear strength and interfacial toughness of the adhesive films at different pressing time. **d** Storage modulus and loss modulus of the adhesive film at different immersion time recorded in the amplitude sweep tests (0.01–100% strain, left) and frequency sweep tests (0.1–10 Hz, right). **e** The relationship between shear strength, interfacial toughness, and the thickness of the adhesive films.

**f** Schematics of reciprocal stretching test. **g** The retention time of the monomer adhered to the tissue in the reciprocal stretching test with different frequencies. The adhesive films are prepared with a ratio of 1: 2 (Carbopol and HPMC-Poloxamer mixture) for (**c**, **d**, **e**, and **g**). In all adhesive films, HPMC-Poloxamer mixture is prepared with the weight ratio of 1 : 1 (HPMC: Poloxamer). The error bars are obtained from 3 trials in each condition. Data are presented as mean values ± SD. Source data are provided as a Source Data file.

tested shear strength and interfacial toughness are presented in Fig. 3b, indicating the adhesion strength of the adhesive films prepared with different weight ratios. In the tests, an adhesive film is placed between two gastric tissues with 1 kPa loaded[49] for a contact time of 1 min. The experimental results demonstrate that the shear strength and interfacial toughness initially increase and then decrease with the increase of the Carbopol content. The highest values of the shear strength and interfacial toughness indicates the maximum adhesion strength of the adhesive film that can be reached, and the values are $7.10 \pm 0.34$ kPa and $29.44 \pm 0.32$ J m$^{-2}$, respectively. A ratio of 1 : 2 (Carbopol: HPMC-Poloxamer mixture) is maintained in this case. The adhesive film is thus prepared with a ratio of 1: 2 (Carbopol: HPMC-Poloxamer mixture) in the subsequent experiments unless otherwise specified.

The hydration state of the adhesive films can influence their adhesion strength, primarily due to the porous structure of the adhesive film (Supplementary Fig. 2). Pressing an adhesive film on wet gastric tissue, the relationship between the adhesion strength and the pressing time is shown in Fig. 3c. The shear strength and interfacial toughness initially increase and then decrease with the increasing pressing time, reaching their highest values at 1 min. The cohesive strength of the adhesive films with different hydration states is then investigated using rheological tests (Methods). The maximum storage

modulus (G′) of the adhesive film is significantly decreased with the increase of immersion time (Fig. 3d). As the immersion time increases from 1 min to 1 h, the storage modulus (G′) is decreased from 36.534 kPa to 16.766 kPa in amplitude sweep tests (0.01–100 % strain) and from 56.947 kPa to 26.686 kPa in frequency sweep tests (0.1–10 Hz), respectively. Meanwhile, the storage modulus (G′) is higher than the loss storage (G″) with the same immerse time, indicating that the cross-linked network formed by hydrogen bonds of the adhesive film is maintained.

Figure 3e presents the relationship between the shear strength and the thickness of the adhesive films, and the relationship between the interfacial toughness and film thickness, in order to testify the adhesion strength of the adhesive films with different thicknesses. Positive correlation between thickness and shear strength, and that between thickness and interfacial toughness are observed. The values of the shear strength and the interfacial toughness reach a plateau at the thickness of 70 μm, i.e., $7.10 \pm 0.34$ kPa and $29.44 \pm 0.32$ J m$^{-2}$, respectively. To further investigate the influence of the film thickness on the adhesion performance, the retention time is characterized using a reciprocal stretching motion test, as schematically shown in Fig. 3f (Methods). In this test, a side layer of the robot, i.e., a monomer, with different thicknesses of the adhesive film is adhered to a piece of gastric

tissue, and they are submerged in SGF. Clamps are used to fix both ends of the gastric tissue, and one side of the tissue is actuated to reciprocally stretch between position I and position II. The right clamp is fixed at position III. Five groups of monomers are prepared, and the retention time is evaluated under the motion frequency from 0.25 Hz to 1.25 Hz. The reciprocal motion frequency of 0.25 Hz exceeds the gastric peristalsis frequency of the human stomach, i.e., approximately 0.05 Hz[50,51]. The retention time is longer than 25 h for all monomers under the motion frequency of 0.25 Hz, and the retention time decreases with the increase of the motion frequency (Fig. 3g), because the increased motion frequency could potentially damage the mechanical structure of the adhesive film more quickly. For instance, the retention time decreases from $26.098 \pm 7.91$ h to $4.194 \pm 0.434$ h as the frequency increases from 0.25 Hz to 1.25 Hz when a monomer with a 70 μm-thick adhesive film is applied. The adhesive film used in this work thus exhibits commendable adhesion performance, demonstrating its potential to serve as a bioadhesive for targeted adhesion.

## Magnetic actuation of the robot

The robot can be accurately actuated by external magnetic fields using both magnetic torques and forces. The schematics of the translational motion of the robot on a gastric tissue is demonstrated in Fig. 4a. The robot performs translational motion at the speed of $v_R$ actuated by applying an external magnetic field gradient. Simulation and analysis of the forces exerted on the robot for its translational motion on gastric tissue are shown and discussed (Fig. 4b and Supplementary Note 1). Under the actuation of the magnetic force $F_m$, the robot realizes translational motion overcoming the friction force $F_f$ between the robot and tissues (Supplementary Fig. 3). Hence, the robot can achieve translational motion once the superposed external force $F_d$ along horizontal direction satisfy the criterion:

$$F_d = F_m^x - F_f > 0 \qquad (1)$$

where $F_m$ is inversely proportional to $d^3$ (Supplementary Note 1). To evaluate the influence of vertical distance on the translational motion

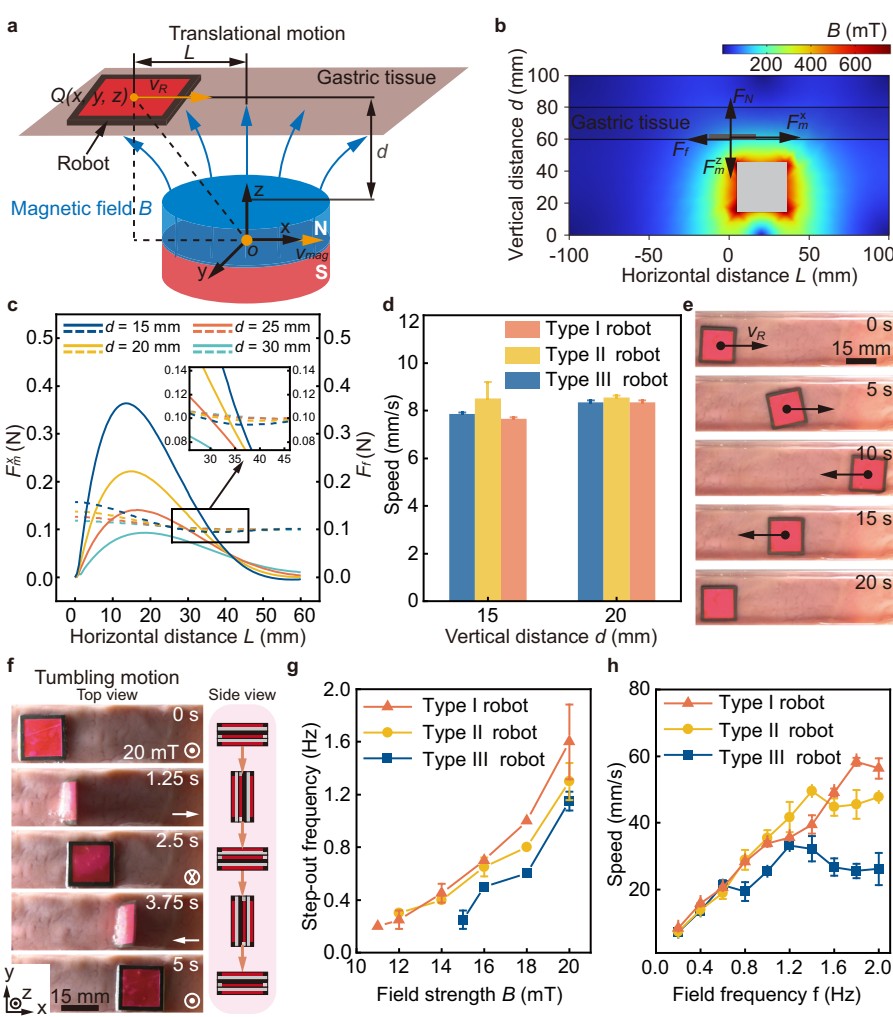

**Fig. 4 | Locomotion modes and performance of the robot. a** Schematics of translational motion of the robot actuated by a magnet. A cartesian coordinate system is defined with the center of the magnet as the origin $O$. Point $Q(x, y, z)$ refers to the center of the robot. The vertical distance between the adjacent surfaces of the magnet and the robot is $d$. The horizontal distance between the centers of the robot and the magnet is $L$. The orange arrows represent the motion direction of the robot and the magnet. **b** Schematic illustration of the forces exerted on the robot performing translational motion on the gastric tissue. Simulation of the magnetic fields generated a magnet ($d = 15$ mm). **c** Relationship between the simulated superposed external force exerted on the robot (Type I robot) performing translational motion and the horizontal distance $L$ at different vertical distances $d$. The solid lines represent the magnetic force $F_m^x$, and the dashed lines represent the friction force $F_f$. **d** Translational speed of the robot when a magnet moves at $^1$0 mm s$^{-1}$ at different vertical distances. **e** Translational motion of the robot on the gastric tissue at a vertical distance $d = 15$ mm. **f** Tumbling motion of the robot on gastric tissue. **g** Relationship between the step-out frequency of the robot performing tumbling motion and the magnetic field strength. **h** Relationship between the speed of the robot performing tumbling motion and the frequency of the magnetic field. The error bars are obtained from 3 trials in each condition. Data are presented as mean values ± SD. Source data are provided as a Source Data file.

of the robot, the superposed external force exerted on the robot (Type I robot) along the horizontal direction is modeled and simulated (Fig. 4c). At $d$ = 15 mm, the superposed external force exerted on the robot exceeds zero when the horizontal distance $L$ between the centers of the robot and the magnet, changes from 3.5 mm to 36 mm, and thus the robot can achieve translational motion. The theoretical analysis also demonstrates that the range of the horizontal distance that the robot can achieve translational motion reduces with increasing the vertical distance (Supplementary Table S1). When the vertical distance $d$ is 30 mm, the superposed external force is zero so the robot remains static (Fig. 4c and Supplementary Fig. 4). Moreover, the translational speed on the gastric tissue of the robot driven by a magnet with a speed of 10 mm s$^{-1}$ is characterized (Fig. 4d). The translational motion of the Type I robot on the gastric tissue is shown in Fig. 4e. The translational motion of the Type II and Type III robots are presented in Supplementary Fig. 5a, b.

The tumbling motion of the Type I robot under a rotating magnetic field with a strength of 20 mT and a frequency of 0.2 Hz is shown in Fig. 4f. The tumbling motion of the other two types of robots is shown in Supplementary Fig. 5c, d. The forces and torques exerted on the robot when it performs tumbling motion are analyzed in Supplementary Fig. 6 and Supplementary Note 2. The robot can achieve stable tumbling motion when the applied field rotation frequency keeps lower than its step-out frequency, which can be expressed as:

$$\varphi_{step-out} = \frac{mB}{c_f} \sin(2\pi f t - \varphi) \qquad (2)$$

Here, $C_f$ is the tumbling damping, $f$ is the rotation frequency of the magnetic field, and $\varphi$ is the angular displacement of the robot. The step-out frequency and translational speed of the robot performing tumbling motion are characterized in Fig. 4g, h, respectively. The experimental results indicate a positive correlation between the step-out frequency of the tumbling motion and the applied field strength (Fig. 4g). Meanwhile, the robot with more layers can reach a higher step-out frequency at the same field strength ($B$ > 15 mT). Moreover, the translational speed of the robot performing tumbling motion increases with the applied field frequency initially, and then decreases after the field frequency reaches the step-out frequency (Fig. 4h), because the robot cannot achieve synchronized motion with the rotating magnetic field.

## On-demand separation by flipping motion

On-demand adhesion and separation of the robot is essential for multi-target adhesion. The influence of the applied field strength on a soft magnetic film is modeled and simulated, in order to investigate the robot flipping and separation process. The magnetic field strength is increased from 0 mT to 50 mT, while the magnetic field direction, defined as the angle between the magnetic field and the y-axis, is fixed as 30° (Fig. 5a and Supplementary Fig. 7a). As the magnetic torque exerted on the magnetic film gradually increases, the robot layer bends and separates from the adhered side layer until its net magnetization align with the direction of the external magnetic field. The simulation results indicate that a magnetic field with a strength of 50 mT is sufficient to perform the separation of the robot. The separation process is then simulated by changing magnetic field direction and keeping the magnetic field strength unchanged (Fig. 5b and Supplementary Fig. 7b). When the magnetic field strength is maintained at 50 mT and the magnetic field direction changes from 0° to 90°, the robot flips around the contact point in response to the magnetic field. The simulated stress distribution shown in the color maps of Fig. 5a, b indicates that, increasing the magnetic field strength and magnetic field direction can result in large deformation of the robot, contributing to the flipping and separation process. The relationship between the stress, the magnetic field strength, and the magnetic field

direction is present in Fig. 5c, showing that stress increases with the field strength and the field direction. The relationship between the layer-layer forces, torque, and the magnetic field direction during separation is present in Supplementary Fig. 7c, showing that horizontal force increases with the field direction, while the vertical force and magnetic torque gradually decrease.

The separation of the robots is performed in two steps, as shown in the schematics in Fig. 5d, including the separation of the Type II robot from the Type I robot and the further separation of the Type II robot. The experimental results of the proposed separation strategy on wet gastric tissue are shown in Fig. 5d and Supplementary Movie 1. A magnet is placed underneath the gastric tissue at a vertical distance of 15 mm to manipulate the robot. Initially, the adhesion between the side layer of the robot and the tissue is formed through applying a vertical magnetic attraction force for approximately 2 min. The first flip motion of a Type II robot and separation of a Type I robot is realized by reversing the magnetic field direction, leaving a side layer of the robot adhered on the tissue ($t$ = 95 ms–255 ms). The separation of the Type II robot is realized using the same actuation strategy, resulting in the second adhered side layer and a free Type III robot ($t$ = 285 ms–315 ms).

## Multi-target adhesion for ex-vivo gastric ulcers

The multi-layer structure and on-demand separation strategy make the robot a promising platform for multi-target adhesion to gastric ulcers. The schematic workflow of the multi-target adhesion using the robot is firstly shown in Fig. 6a. A Type I robot contacts the gastric tissue and is navigated to cover the gastric ulcer through translational motion. Adhesion will be formed between the side layer of the robot and the gastric tissue. A Type II robot separates from the adhered side layer through a flipping motion, and it is subsequently navigated to the next ulcer for coverage and adhesion. The separation of the Type II robot is realized using the same strategy, and the generated Type III robot can be used for the third ulcer adhesion through a flipping motion.

The ex-vivo experiment of multi-target adhesion using the robot is conducted on porcine gastric tissue (Fig. 6b and Supplementary Movie 2). Three artificial ulcers are created, i.e., Target I - III. A Type I robot is delivered to Target I actuated by an external magnetic field ($t$ = 26 s), and the adhesion between the side layer of the robot and the gastric tissue is formed by applying a vertical magnetic attraction force for approximately 2 min. Subsequently, a Type II robot separates from the adhered side layer by changing the direction of the magnetic field ($t$ = 144 s) and is navigated to Target II to adhere to it ($t$ = 203 s). Finally, a Type III robot separates from the second adhered side layer ($t$ = 729 s) and moves to Target III for the third adhesion ($t$ = 776 s). Multiple adhesion using the robot can also be applied to cover the larger ulcer. This method can enlarge the total coverable region of the robot by implementing multiple adhesion adjacently. Herein, we demonstrate that a long ulcer with a length of 35 mm and a width of 5 mm can be covered by the robot as shown in Supplementary Fig. 8. During the second delivery and adhesion, a longer indwelling time is required for the side layer of the robot to form adhesion with the tissue, which could be attributed to the mucus adhered to the robot during the navigation and delivery process. The experimental results on the time required for adhesion are obtained from the trials on the gastric tissue (Supplementary Fig. 9). The time required for the first and second adhesion are 184 ± 102 s and 382 ± 164 s, respectively. The gastric tissue with three robot layers adhered is then immersed in SGF to evaluate the retention performance of the adhesive film on the ex-vivo tissue. After 12 h, the adhesion of the adhesive film still remains (Supplementary Fig. 10).

Moreover, the ex-vivo experiment of multi-target adhesion using the robot is performed inside a porcine stomach with artificial ulcers, and ultrasound imaging is used for navigation (Fig. 7a and Supplementary Movie 3). The features of the ulcers and the robots in ultrasound images are shown in Supplementary Fig. 11. The setup of the

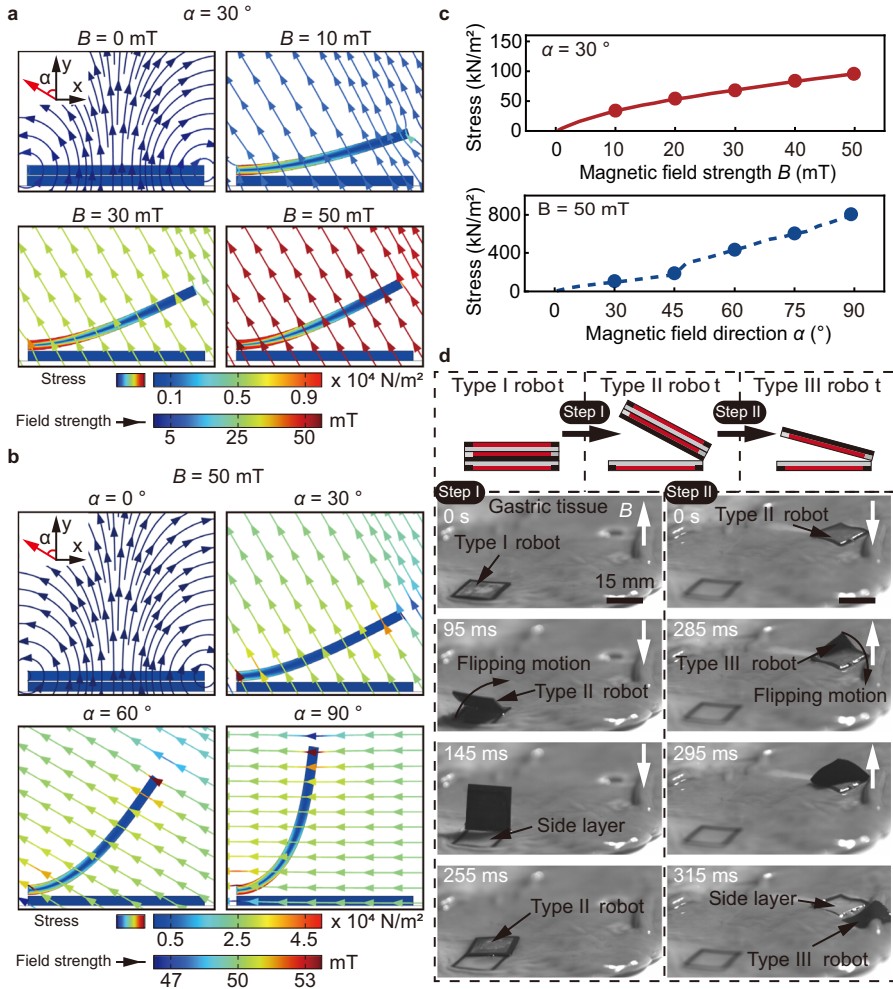

**Fig. 5 | On-demand separation of the robot. a** Simulation of the separation process of a soft magnetic film actuated by a magnetic field with increased field strength (from 0 mT to 50 mT) and constant magnetic field direction. **b** Simulation of the separation process of a soft magnetic film actuated by a magnetic field with changed field direction (from 0° to 90°) and constant magnetic field strength. **c** The simulation results of the stress of the film with magnetic field strength and direction. **d** Schematics and experimental demonstration of the separation strategy, from the Type I robot to the Type II robot (Step I, left), and from the Type II robot to the Type III robot (Step II, right). Source data are provided as a Source Data file.

experiment is shown in Supplementary Fig. 12 (Methods). In the ultrasound images, the ulcer (the blue dashed ellipse, Fig. 7a) harms and breaks the gastric mucosa (the white dashed line in Fig. 7a, $t = 0$ s). The robots are labeled using red dashed rectangles. A Type I robot is placed into the stomach through the esophagus and tracked using ultrasound imaging ($t = 0$ s). The robot contacts the gastric mucosa and is navigated to cover the ulcer I ($t = 31$ s). Adhesion is formed through exerting a magnetic stressing force on the robot towards the mucosa for approximately 5 min. A Type II robot separates from the first adhered side layer through a flipping motion by reversing the magnetic field direction, as shown by the separation of two white patterns labeled by the red dashed rectangles ($t = 340$ s–342 s). The navigation-adhesion process of a Type II robot to ulcer II is subsequently demonstrated ($t = 342$ s–372 s). Finally, a Type III robot separates from the second adhered side layer and is navigated to ulcer III to complete the third adhesion ($t = 1016$ s–1040 s).

A postoperative ultrasound scan is performed on the three ulcers to evaluate the results of multi-target adhesion (Fig. 7b). The ultrasound images indicate that the three layers of the robot successfully cover and adhere to the three ulcers. The stomach is then dissected, and a straightforward observation of the ulcer spots is made (Fig. 7c). All ulcers are covered with the robot layers, including the one locates near the folds of the stomach and the one locates in the confined space

between two folds. The schematics inset of in Fig. 7c shows the initial positions of the ulcers in the porcine stomach.

A quantitative analysis of the biocompatibility of the robot is then conducted (Fig. 7d and Supplementary Fig. 13). In-vitro tests are conducted by separately co-culturing the adhesive film and the soft magnetic substrate of the robot with human gastric mucosal epithelial cells (GES-1) for 24 h (Methods). The results show that the viabilities of the cells co-cultured with the adhesive film and the soft magnetic substrate are 99.0% and 97.5%, respectively, and they are compared to the control group with a cell viability of 97.3%.

## Multi-target adhesion in an in-vivo porcine stomach

In-vivo stomachs add challenges and complexities compared with ex-vivo ones. While the in-vivo stomach and the ex-vivo stomach are structurally similar, the internal environment of the living porcine stomach brings significant complexities, such as the dynamic peristalsis and the consistent secretion of gastric mucus. The peristaltic motion of the living stomach poses challenges in monitoring and precisely locating the robot within the gastric environment. Additionally, the continuous secretion of gastric mucus makes effective contact difficult between the adhesive film and the tissue, which could prevent the formation of stable adhesion. Furthermore, the in-vivo stomach has more folds on the surface. Therefore, in-vivo experiments

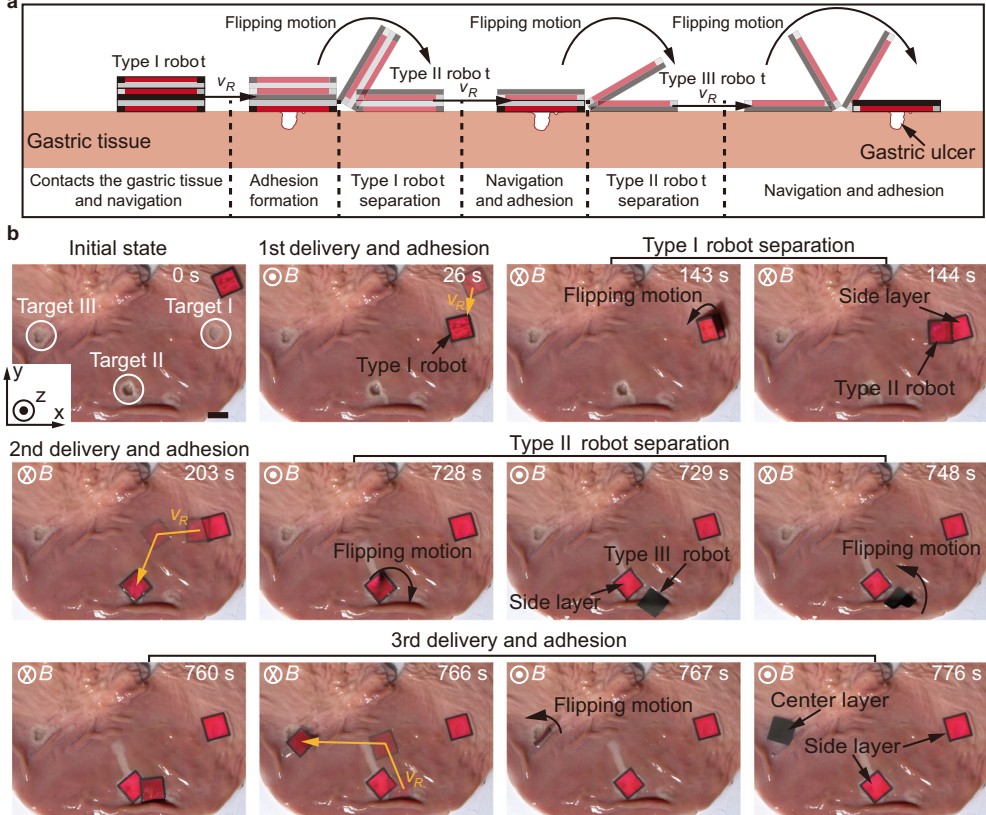

**Fig. 6 | Demonstration of multi-target adhesion on ex-vivo gastric tissue. a** Schematics of the multi-target adhesion on the gastric tissue. **b** Experimental results of the multi-target adhesion using the robot on ex-vivo gastric tissue. The scale bar is 15 mm.

are necessary. The multi-target adhesion using the robot in an in-vivo porcine stomach is performed (Fig. 8 and Supplementary Movie 4). The schematics of the in-vivo experiment is demonstrated in Fig. 8a (Methods). Using real-time ultrasound imaging for tracking, the robot is capable of approaching, covering and adhering to the ulcers actuated by magnetic field (Fig. 8b). The side layer adheres to the surrounding tissue of ulcer I, and then the Type II robot separates from it. The Type II robot and the Type III robot separated from it successfully cover and adhere to ulcer II and ulcer III, respectively, based on the ultrasound imaging feedback. The stomach is then dissected to evaluate the accuracy of the adhered layer of the robot. It is observed that the layers of the robot adhere to three positions on the folded gastric mucosa (Fig. 8c), and after removing the layers, all the ulcers are revealed (Fig. 8d). The experimental results demonstrate that the proposed robot can tackle the challenge and achieve multi-target adhesion with high precision inside an in-vivo stomach with mucus.

## Discussion

In summary, we have developed a magnetic soft robot consisting of multiple layers with distinct in-plane structures and functions to achieve on-demand adhesion to different lesions. Designed materials interface and tailored interlayer interactions are taken into consideration. The robot consists of three layers, including one center layer and two side layers. Each layer is composed of a soft magnetic substrate and an embedded adhesive film. To achieve adhesion to wet tissue, we prepare the Carbopol-based adhesive film that can adhere to the tissue through hydrogen bonds. The adhesion performance of the adhesive film is evaluated and optimized. Furthermore, the interlayer interaction of the robot is adjusted by designing each layer with a different substrate. The side layer consists of a nonmagnetic base and a magnetic frame, while the center layer consists of a magnetic base and a nonmagnetic frame. The three layers are magnetized in the same

direction perpendicular to their plane, providing interlayer magnetic attractions. Once the side layer adheres to the wet tissue, the magnetic torque can be used to flip the robot and separate it from the adhered side layer. By leveraging the on-demand adhesion and separation strategy, we validate the implementation of the robot for multi-target adhesion in ex-vivo and in-vivo environments, demonstrating its potential in treating gastric ulcers. In summary, by integrating functional materials into soft robots, more possibilities could be provided to broaden the applications.

Considering the specific needs required by a magnetic field actuation system for medical scenarios, the effective operation distance of the robot can be extended by increasing magnetic field strength and gradient. For systems integrated with permanent magnet, increasing the volume of the magnet can enhance both the magnetic field strength and gradient, while for electromagnetic coils, boosting the current amplitude in the electromagnetic coil and implementing water-cooling equipment could also be a promising method. For medical imaging targeting deep regions, we can apply this strategy under the guidance of X-ray imaging. Oral-taking barium meal can locate the ulcers[52], and the robot can be navigated to cover and adhere to them. It is noted that a robot arm with a permanent magnet serving as its end effector can also realize the on-demand actuation of the robots (Supplementary Fig. 14). In addition, loading drugs into the adhesive film of the robot will help further transform the technology into clinical applications that maintain its own adhesion performance and movement (Supplementary Figs. 15 and 16).

## Methods
### Materials
Carbopol 971 P is obtained from CHINE INTERNATIONAL (Shanghai, China). Poloxamer 407 is purchased from Solarbio LIFE SCIENCES (Beijing, China). Hydroxypropylmethylcellulose (HPMC) is purchased

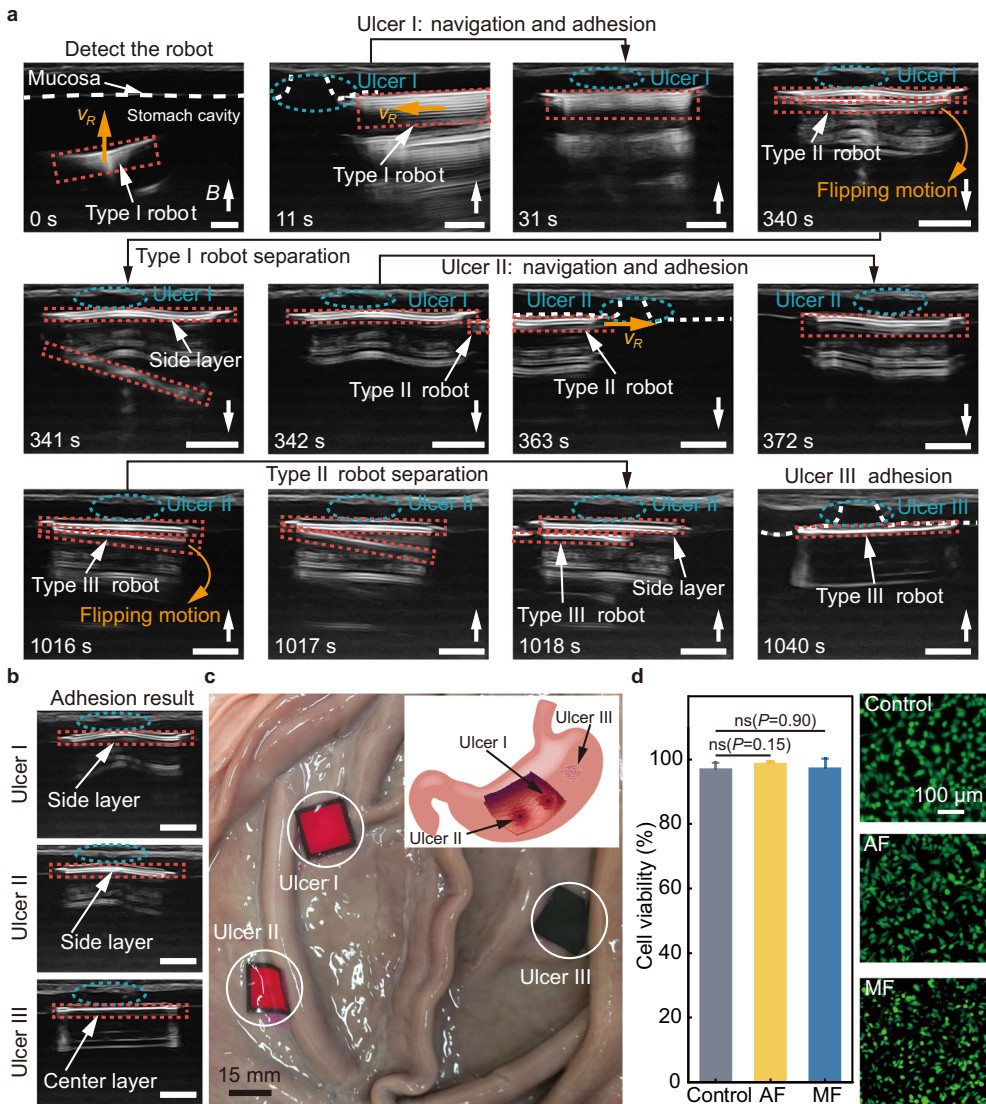

**Fig. 7 | Multiple-target adhesion in an ex-vivo porcine stomach. a** The multi-target adhesion using the robot in an ex-vivo porcine stomach filled with liquid under real-time tracking of ultrasound imaging. The red dashed rectangles, the white dashed lines and the blue dashed ellipses represent the robot, the gastric mucosa and the ulcer, respectively. The orange arrows represent the translational motion direction of the robot. The scale bar is 5 mm. **b** Postoperative check with ultrasound imaging. The scale bar is 5 mm. **c** The dissected porcine stomach after the multi-target adhesion. The schematics illustrates the initial positions of the ulcers in the porcine stomach. **d** The cell viability (left) of human gastric mucosal epithelial cells (GES-1) in three groups, i.e., control group, adhesive film (AF) group, and soft magnetic substrate (MF) group. For the control group, nothing is added to the culture dish. The adhesive film and the soft magnetic substrate are co-cultured with the cells for the experimental groups, respectively. Live/dead staining image of GES-1 cells in three treatment groups after 24-h co-culture (Right). Data are presented as mean values ± SD. Source data are provided as a Source Data file.

by MACKLIN (Shanghai, China). All other chemicals are of extra pure reagent grade and are used as received.

## Fabrication of magnetic multi-layer soft robot

As shown in Supplementary Fig. 1a, the magnetic film is prepared by spin-coating magnetic slurry, which is prepared by mixing hard-magnetic neodymium-iron-boron (NdFeB) microparticles and poly-dimethylsiloxane (PDMS) in a mass ratio of 2: 1.

The adhesive film is prepared from Carbopol, Poloxamer, and HPMC. The Carbopol, HPMC, and Poloxamer are dissolved in an ethanol/water mixture (50/50: v/v). Rhodamine B is physically mixed into the solution to prepare the pink film for better visualization. The obtained pre-solution is poured into the petri dish, and the adhesive film is obtained by evaporating the remaining solution with heat.

Each component of the layer is obtained from different films by laser cutting and assembled into side and center layers respectively after plasma treatment (Supplementary Fig. 1a, b). The layers are programmed with the same magnetization direction perpendicular to their plane under a pulsed magnetic field (Supplementary Fig. 1c). Last, the robot is assembled under interlayer magnetic attraction. Detailed information of the robots is shown in Supplementary Table S2.

## Mechanical tests

The fracture toughness of the adhesive films prepared with different weight ratios of the Carbopol and HPMC-Poloxamer mixture is measured using tensile tests of thin rectangular samples (20 mm in length, 15 mm in width) with a mechanical testing machine (0.1 kN load-cell). All tests are conducted with a constant tensile speed of 50 mm min$^{-1}$. The fracture toughness of the adhesive film is calculated using a reported method.

Rheological tests are performed using a rheometer (Anton Paar MCR302e, Austria) with rheocompass software. The dynamic moduli

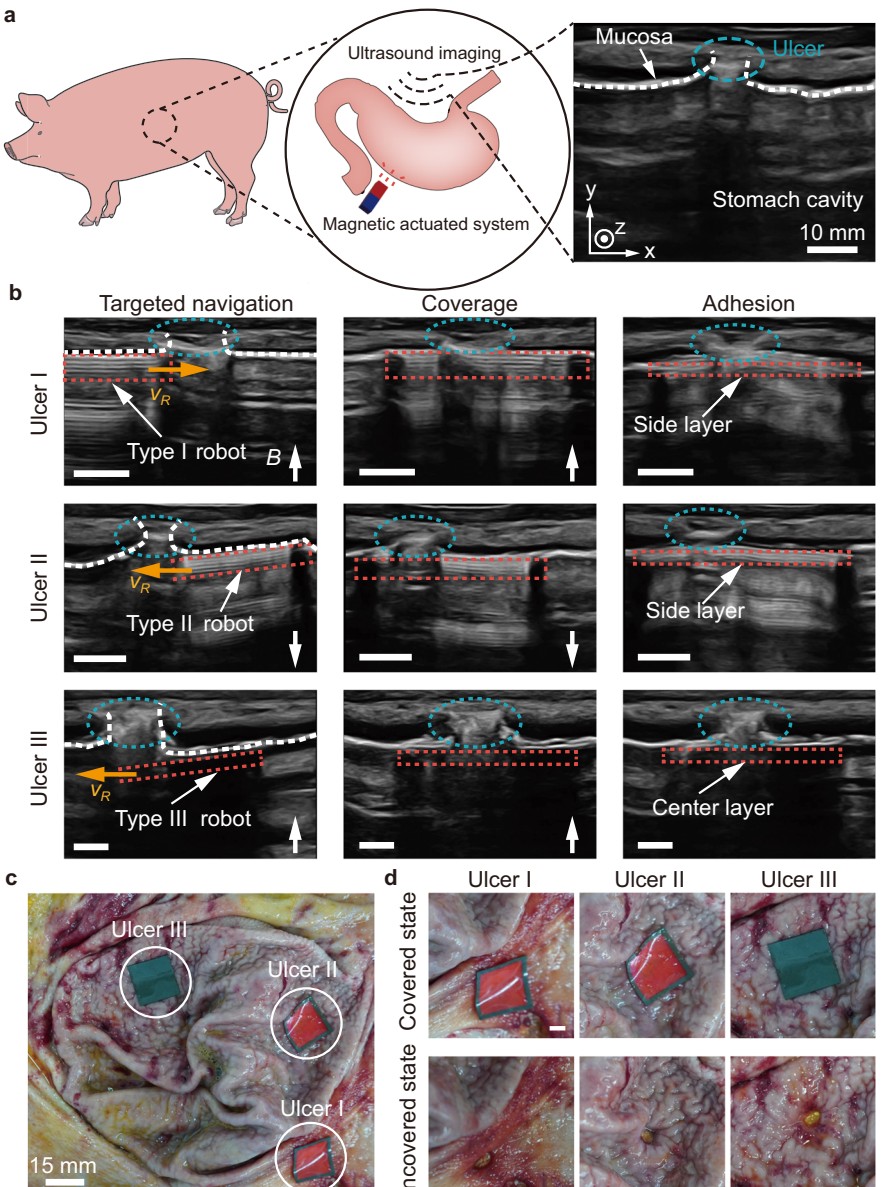

**Fig. 8 | On-demand targeted adhesion in an in-vivo porcine stomach.**
**a** Schematics of the experiment. The white dashed line and the blue dashed ellipses represent the gastric mucosa and the ulcer, respectively. **b** Real-time navigation of a robot in an in-vivo porcine stomach after gastric emptying for multi-target adhesion. The steps include targeted navigation, coverage, and adhesion. The red dashed rectangles represent the robot. The orange arrows represent the translational motion direction of the robot. The scale bar is 5 mm. **c** The dissected porcine stomach after the multi-target adhesion. **d** Detailed views of the covered ulcers and those revealed after peeling the robot layers. The scale bar is 5 mm. Panel (**a**) is partly generated using Servier Medical Art, provided by Servier, licensed under a Creative Commons Attribution 3.0 unported license.

(Storage modulus, G′ and Loss modulus, G″) of the adhesive films are obtained. Amplitude sweep tests are conducted by varying the strain from 0.01 to 100% with a frequency of 1 Hz. Frequency sweep tests are conducted by varying the frequency from 0.1 to 10 Hz with a strain of 1%.

**In vitro adhesion tests**
Quantitative tissue adhesion strength tests, including the Lap-shear and T-peel tests, are performed according to the ASTM F2255 and F2256. The pig gastric tissue is used as biological tissue material. The two tissues are placed into a universal testing machine (Instron, America) for tensile loading at a strain rate of 50 mm min⁻¹. The adhesive film is placed between the tissues and pressed under 1 kPa for different time.

To measure shear strength, adhered samples (the adhesive film) with an adhesion area of width 15 mm and length 15 mm are prepared

and tested by the standard lap-shear test (ASTM F2255) with a mechanical testing machine (0.1 kN load-cell). Shear strength is determined by dividing the maximum force by the adhesion area.

To measure interfacial toughness, adhered samples (the adhesive film) with a width of 15 mm are prepared and tested by the standard 180-degree peel test (ASTM F2256) using a mechanical testing machine (0.1 kN load-cell). The measured force reaches a plateau as the peeling process enters the steady state. Interfacial toughness is determined by dividing two times the plateau force (for a 180-degree peel test) by the width of the tissue sample.

To characterize the retention time of the adhesive film under cyclic loading, porcine gastric tissue adhered with a monomer is immersed in SGF. The stretching equipment (KeyFactor, China) stretches the pig gastric tissue at different frequencies (0.25, 0.5, 0.75, 1, 1.25 Hz) and records the retention time of the monomer.

### In vitro biocompatibility test

To evaluate in-vitro biocompatibility and cytotoxicity of the adhesive layer, LIVE/DEAD assay is used to assess Human gastric epithelial cell line (GES−1, Xiamen Immocell Biotechnology Co.,Ltd.). Cut the adhesive film to $2 \times 2$ mm$^2$ and co-culture it with GES−1, without adding the adhesive film as a control. GES-1 cells are seeded in 96-well plates at a density of $0.5 \times 10^5$ cells ($n = 3$ per each group). The cells are then incubated at 37 °C for 24 h in 5% $CO_2$ atmosphere. The cell viability is determined by a LIVE/DEAD viability/cytotoxicity kit for mammalian cells (Thermo Fisher Scientific) by adding 4 M calcein and ethidium homodimer-1 into the culture media. A microscope (Nikon ECLIPSE Ti2) is used to image live cells with excitation/emission at 495 nm/515 nm and dead cells at 495 nm/635 nm, respectively. The cell viability is calculated by counting live (green fluorescence) and dead (red fluorescence) cells by using ImageJ (version 2.1.0).

### Ultrasound imaging for ex-vivo and in-vivo tests

In this study, ultrasound imaging is employed to track the robots and the ulcers. Under ultrasound images, the mucosa appears as a continuous white line in the water-filling environment. An obvious change in the contour can be observed at the site of the ulcer. In environments with air, the ultrasound images show the presence of irregular white signals, which can serve as an indication of the location of ulcers. In a stomach filled with fluid or in an empty stomach, the location and pattern of the robot are identified by tracking the stacked white lines in the ultrasound images. The thickness of the stack decreases when a Type I robot transits into a Type III robot. In particular, the location where the white signal (the yellow dashed rectangles, Supplementary Fig. 11) disappears can be used to locate the Type III robot in the air environment, allowing for tracking of the Type III robot as it moves in an in-vivo porcine stomach after gastric emptying.

### Experimental procedure of ex-vivo and in-vivo tests

Experimental procedures of ex-vivo and in-vivo experiments can be listed as follows. Two operators collaborate to achieve multi-target adhesion of the robot. Operator 1 operates the ultrasound transducer to locate the robot. Once the robot is located, Operator 2 manipulates the robot using a cylindrical magnet. Operators 1 and 2 move the transducer and magnet synchronously to actuate the robot toward and then cover target I. Subsequently, once the adhesion between the adhesive film and the tissue is formed under magnetic attraction. Operator 2 then reverses the magnet to separate Type II robot from the adhered side layer. This systematic approach is also applied to cover targets II and III.

### Preparation of porcine stomach for ex-vivo tests

All ex-vivo pig stomachs are purchased from Taobao, China. The organs are emptied and minimally cleaned by water without damaging the mucosa. Acetic acid-induced gastric ulcers are established using a previously reported protocol[42]. Use surgical scissors to cut out three areas of 100 mm$^2$ on the surface of the pig stomach, and 75% acetic acid is injected into the incision and maintained for 120 s. For ex-vivo porcine stomach, three artificial ulcers are created on different sides of the stomach. For ex-vivo tests, the dimensions of the robot (Type I robot) are $18 \times 18 \times 1.5$ mm$^3$ and the weight is $0.612 \pm 0.054$ g.

### In-vivo tests in a pig

After the appropriate depth of anesthesia and analgesia, the stomach is exposed by surgical laparotomy. A small incision is made in the esophagus using a surgical knife to remove the stomach contents. The stomach is then washed with physiological saline. A surgical scissor is used to create three ulcers at the gastric mucosa, each about 1 cm in diameter, and the location of the ulcers is determined using ultrasound imaging. The robot is delivered into the stomach through the incision and navigated using ultrasound. The direction and distance of the

magnetic field are adjusted to allow the robot to form adhesion with the tissue and separate from the side layer. The final adhesion result is examined using ultrasound imaging. Under deep anesthesia, the animal is then euthanized by intravenous injection of potassium chloride, and the stomach is dissected to observe the final adhesive result. For in-vivo tests, the dimensions of the robot (Type I robot) are $18 \times 18 \times 1.5$ mm$^3$ and the weight is $0.612 \pm 0.054$ g.

### Simulation

Simulation of the stress distribution when the robot performed translational motion on the stomach and the separation between a soft magnetic film and the adhered side layer to the tissue under magnetic fields with different directions and strength used the commercial software Comsol. Young's modulus $G = 3.2$ MPa and magnetization $M = 152.076$ kA m$^{-1}$ measured from the experiment are used as the input parameters.

### Statistical analysis

The data obtained in this study are all reported as mean ± standard deviation (SD) from at least three separate experiments. Values from two groups are compared using Student's $t$ test. A $P$ value of 0.05 or less is considered to be statistically significant. $^*P < 0.05$, $^{**}P < 0.01$, $^{***}P < 0.001$, $^{****}P < 0.0001$; ns, not significant.

### Reporting summary

Further information on research design is available in the Nature Portfolio Reporting Summary linked to this article.

## Data availability

The data generated in this study are provided in the Supplementary Information/Source Data file. Additional data are available from the corresponding author on request. Source data for the figures are provided with this paper. Source data are provided with this paper.

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

## Acknowledgements

We thank Kaiwen Fang and Guoqing Chen for their assistance in animal surgeries. This work was financially supported by the National Key R&D Program of China under Project No. 2022YFA1207100, the National Natural Science Foundation of China under Project No. 62103347, No. 62073208 and No. 61933008, the Innovation Program of Shanghai Municipal Education Commission under Grant 2021-01-07-00-09-E0013, Shenzhen Science and Technology Program under Grant No. RCBS20210609103155061, Guangdong Basic and Applied Basic Research Foundation under Project No. 2022A1515110499, and the Shenzhen Institute of Artificial Intelligence and Robotics for Society under Project No. AC01202101109, Shanghai Science and Technology plan project under Grant 23ZR1422300.

## Author contributions

Z.C., Y.W., N.L. and J.Y. conceived the study, designed the experiments. Z.C., Y.W., J.L., N.L. and J.Y. wrote the manuscript. Z.C., Y.W. and H.C. performed the experiments and analyzed data. Z.C. and Y.W. provided conceptual and technical input for the numerical simulations. Z.C., Y.W., F.D. and H.C. assisted with the experiments of clinical imaging. H.P., S.X., Y.S., N.L. and J.Y. assisted with the supervision of experiments and edited the manuscript. N.L. and J.Y. supervised the project. All authors discussed the results and reviewed the manuscript.

## Competing interests

The authors declare no competing interests.
