## [Peer Review File · Nature Communications]

REVIEWER COMMENTS

Reviewer #1 (Remarks to the Author):

In the manuscript by Chen et al., the authors introduce an innovative multi-layered magnetic robotic adhesive designed specifically for the minimally invasive treatment of gastric ulcers. The adhesive material is formulated from crosslinked Carbopol, Poloxamer, and HPMC, while the magnetic component is a composite of neodymium iron boron (a hard magnetic material) and PDMS.

The thorough characterization of both the materials used and the final robot assembly is commendable. Of particular importance is the characterization of adhesion in the presence of simulated gastric fluid (SGF) over extended durations. The manuscript is articulately written and appears comprehensive.

However, I do have a few points that require clarification:

Regarding the ulcer size and number: Could the authors provide a rationale for choosing this specific ulcer size and explain how the robot adhesive might be adapted for larger ulcers, especially those with uneven or rough surfaces?

Concerning the ultrasound images: There are instances where lines obscure the actual signals. In particular, the signal for the Type III robot, Fig 8 seems faint or sometimes even absent. Also the ulcer seems hard to identify on US. Could this be elaborated upon?

Pertaining to the in vivo study: Based on the photos, it appears some adhesives either delaminated or did not achieve optimal tissue contact. Could the authors shed light on how this might influence adhesion efficacy?

Could the authors elaborate on the limitations related to external control (distance, DoF) and the feasibility of translating this technology into clinical settings?

Reviewer #2 (Remarks to the Author):

The authors present a multi-layered soft robot where each layer is connected to another by means of magnetic forces. Additionally, each layer contains an adhesive film consisting of crosslinked Carbopol, Poloxamer, and HPMC that is able to form hydrogen bonds with the stomach lining. Movement and post-adhesion layer detachment of the robot is achieved by the application of magnetic fields. In order to achieve adhesion, magnetic forces exerted by an external magnet are used to press down the film on the stomach lining. The story of the manuscript is compelling and the graphic material including figures and videos are of a high visual and informative quality. However, I ask the authors to consider the following:

- Regarding the clinical application. The adhesive layer is able to bond with gastric tissue and cover it, but does not appear to promote healing of the ulcer. Is there a specific clinical need for covering an ulcer or can the robot be extended with drugs to promote healing of the ulcer?
- Regarding magnetic actuation. Based on the design of the robot, there appears to be a missing characterization (can also be based on simulation) of the layer-layer interaction forces and torques during separation. In addition, this should be connected to the required magnetic field to achieve separation and placed in context with magnetic fields typically achievable by state-of-the-art medical-purpose magnetic field generating systems (e.g., DOI 10.1002/adfm.202005137).
- Continuing on my previous comment, based on Fig. 4c the magnetic force of 6N exerted on the robot by the external permanent magnet seems high when comparing to other works reporting forces on magnetic polymers by magnets (e.g., DOI 10.1089/soro.2022.0031). I am missing some validation regarding the actual forces exerted by the magnet on the robot, as well as a quantification of the necessary field gradients. In reality, I assume a magnet-robot distance of 15mm is unachievable at all times around the stomach, and any increase in distance has a significant effect on the achievable magnetic forces on the robot.
- The experimental procedure can be elaborated more. Was the ultrasound transducer operated manually as well as the magnet, simultaneously by one person?
- The ex vivo demonstrations are compelling. However, what do the subsequent in vivo experiments aim to show in addition to the ex vivo experiments, considering the animal is euthanized post-experiment and the stomach dissected to again inspect ex vivo? I acknowledge that the authors mention that the in-vivo gastric environment is more complex with secretion of mucus on the gastric mucosa. An ex vivo stomach is structurally similar to an in vivo stomach, hence an elaboration of the added complexity is warranted. Similarly, is the secretion of mucus on the gastric mucosa a concern, in the sense that it may increase friction?
- Based on my understanding of the visual material, it appears the experiments were performed with the gastric mucosa below the robot in the direction of gravity. How will the robot work if an ulcer is located on the upper gastric wall. If the answer is magnetic attraction, what is the weight of the

robot and which field gradients are necessary to achieve adherence against the direction of gravity (based on simple calculations)?

Minor comments:

- Reference 18 contains n/a

Reviewer #3 (Remarks to the Author):

This paper presents a magnetic soft robot consisting of multiple layers with distinct in-plane structures and functions to achieve on-demand adhesion to different lesions.

In addition, it demonstrates two locomotion modes and the separation with one side layer adhered to tissues.

However, the following issues should be considered and addressed in the revised manuscript.

1. If the soft robot is placed on the stomach surface for more than a certain period of time without magnetic force applied to the robot,

does not the adhesion between each layer and the stomach surface occur?

If so, is there any possibility that this will lose the adhesion of the layer?

2. Detailed information such as dimensions, weight, etc. of the proposed robot should be provided.

In addition, the robot is expected to be light because it consists of three thin layers.

So, it is expected that the flow of gastric fluid affects the robot's movement.

3. The proposed robot should be loaded with drugs.

It should also show how efficiently the drug can be loaded, how effectively the drug is released, and the performance of the treatment by the released drug.

You can also load a model drug rather than a real drug.

In addition, since loading drugs can affect the mobility or adhesion of the proposed robot, the related experiments should be included or discussed.

4. It appears that a permanent magnet was used to move the robot.

Is the magnet controlled manually?

Or do you use a macro robot arm for the locomotion of the soft robot?

Also, when the layer is separated by magnetic force, how is the magnetic force controlled at this time?

Finally, the information on how the robot is controlled and the actuation system should be included in the manuscript.

Dear Editor and Reviewers:

We report below a detailed response to the reviewers' reports. The original reviewer comments are in italic and Response appears in blue font. When addressing the reviewers' concerns, we also explicitly listed all the changes we made in the revised manuscript, which are color-coded red. The indexes of the references are listed as they are in the revised manuscript.

Reviewer 1:

In the manuscript by Chen et al., the authors introduce an innovative multi-layered magnetic robotic adhesive designed specifically for the minimally invasive treatment of gastric ulcers. The adhesive material is formulated from crosslinked Carbopol, Poloxamer, and HPMC, while the magnetic component is a composite of neodymium iron boron (a hard magnetic material) and PDMS. The thorough characterization of both the materials used and the final robot assembly is commendable. Of particular importance is the characterization of adhesion in the presence of simulated gastric fluid (SGF) over extended durations. The manuscript is articulately written and appears comprehensive. However, I do have a few points that require clarification:

Response: We thank the reviewer for the constructive comments on our work. We have carefully addressed all the questions raised by the reviewer point-by-point and revised our manuscript accordingly.

Comment #1: *Regarding the ulcer size and number: Could the authors provide a rationale for choosing this specific ulcer size and explain how the robot adhesive might be adapted for larger ulcers, especially those with uneven or rough surfaces?*

Response: We thank the reviewer for the comment. The average size of human benign ulcers from 14,400 patients is 10 mm (Koçak, E. et al., *Wien. Klin. Wochenschr.*, 125, 21-25, 2013). We also prepare the artificial ulcers with the size of 10 mm in this work (S. Okabe, et al., *Biol. Pharm. Bull.*, 28, 1321-1341, 2005). For ulcers with larger sizes, a robot with larger sizes and customized shapes can be prepared to cover and adhere to the ulcers, and scaling up the robot size is achievable using the current fabrication process. Furthermore, multiple adhesion using the multi-layer robot can be adopted to cover the larger ulcer. This method can enlarge the total coverable region of the robot by implementing multiple adhesion adjacently. Herein, we demonstrate that a larger ulcer with a length of 35 mm and a width of 5 mm can be covered by the robot as shown in Fig. R1. A Type I robot moves to one side of the larger ulcer actuated by an external magnetic field ($t = 6$ s), and the adhesion between the side layer of the robot and the gastric tissue is formed by applying a vertical magnetic attraction force for 3 mins. Subsequently, a Type II robot separates from the adhered side layer by changing the direction of the magnetic field ($t = 212$ s) and is navigated to the other side of the larger ulcer to adhere on it ($t = 238$ s). Finally, a Type III robot is separated ($t = 631$ s) and moves to the middle of the larger ulcer for the third adhesion ($t = 699$ s).

Fig. R1 Experimental results of the multiple adhesion of the robot for larger ulcer on ex-vivo gastric tissue. The orange arrows represent the translational motion direction of the robots. The scale bar is 20 mm.

For ulcers with uneven or rough surfaces, the soft robot is able to deform to fit onto uneven surfaces, which is already demonstrated in Fig. 7c of the manuscript. In the figure, all ulcers are covered with the robot layers, including the one located at the large fold (Ulcer II) and the one located in the confined space between two folds (Ulcer III). On surfaces like mucosa covered with mucus fluid, the adhesive film is capable of contacting and forming adhesion with gastric tissue by applying magnetic forces, as shown in Fig. 8c. In summary, the magnetic multi-layer soft robot proposed in this work is capable of achieving effective adhesion of multiple ulcers and larger ulcers in the unstructured environments of the stomach.

To clarify this issue, we have added the results presented in Fig. R1 into the supplementary information, as Supplementary Fig. 9. We have revised the second paragraph in Section **Multi-target adhesion for ex-vivo gastric ulcers** by adding “Multiple adhesion using the robot can also be applied to cover the larger ulcer. This method can enlarge the total coverable region of the robot by implementing multiple adhesion adjacently. Herein, we demonstrate that a long ulcer with a length of 35 mm and a width of 5 mm can be covered by the robot as shown in Supplementary Fig. 9.”.

Comment #2: Concerning the ultrasound images: There are instances where lines obscure the actual signals. In particular, the signal for the Type III robot, Fig 8 seems faint or sometimes even absent. Also the ulcer seems hard to identify on US. Could this be elaborated upon?

Response: Thank you for the comments. The ultrasound image quality depends on the environment since the ultrasound wave is sensitive to the transmitting medium. We characterize the ultrasound imaging of the ulcers on a piece of gastric tissue in different environments. Ulcers with different sizes, i.e., 5 mm and 10 mm in diameter, are prepared by removing the mucosa, as shown in Fig. R2a. To better identify ulcers, we conduct ultrasound scanning experiments on ulcers in environments filled with water and air, respectively, as shown in Fig. R2b. Under ultrasound images, the mucosa appears as a continuous white line in the water-filling environment. An obvious change in the contour can be observed at the site of the ulcer. In environments with air, the ultrasound images show the presence of irregular white signals, which can serve as an indication of the location of ulcers.

What’s more, through real-time monitoring, the image signals of the ulcers (the blue dashed ellipse, Fig. R2c) and the robots (the red dashed rectangles, Fig. R2c) can be distinguished clearly. To comprehensively profile the characteristic ultrasound signal of the ulcers and the robots, we have acquired the ultrasound image of them in environments with water and air as the medium, respectively (Fig. R2c). In the ultrasound images, the presence of stacked white lines provides a clear visual indication of the robot's location and

shape. The stacked white lines are attributed to the entrapped air between the robot layers. Notably, as transition occurs from Type I to Type III robots in liquid environments, the stacked white line diminishes due to the disappearance of inter-layer gaps. This phenomenon is consistent with the ex-vivo experimental results shown in Fig. 7. These results serve as compelling evidence for tracking the motion and features of the robot in a liquid environment.

Additionally, the ultrasound imaging conducted in air demonstrates that Type I and Type II robots can also be recognized precisely by identifying the stacked white lines. As the reviewer mentioned, the Type III robot appears faint in the ultrasound image. Therefore, a new visual indication is employed. We utilize the location where the white signal (the yellow dashed rectangles, Fig. R2c) disappeared to locate its position, which allows us to track the Type III robot as it moves, as demonstrated in Fig. R2c and Fig. 8b. Supplementary Video 4 also illustrates the movement of the Type III robot within a stomach with air. Consequently, the robot can be clearly observed in the stomachs filled with water and air, which further allows the visual tracking and manipulation.

In order to give more details about the ultrasound imaging, we have added the results presented in Fig. R2 into the supplementary information, as Supplementary Fig. 12. We have revised the third paragraph in Section **Multi-target adhesion for ex-vivo gastric ulcers** by adding “**The features of the ulcers and the robots in ultrasound images are shown in Supplementary Fig.12.**”. The captions of Fig.7 and Fig. 8 have also been revised.

A section in **Methods** is also added to clarify the process of performing ultrasound imaging as follows.

Ultrasound imaging for ex-vivo and in-vivo tests

In this study, ultrasound imaging is employed to track the robots and the ulcers. Under ultrasound images, the mucosa appears as a continuous white line in the water-filling environment. An obvious change in the contour can be observed at the site of the ulcer. In environments with air, the ultrasound images show the presence of irregular white signals, which can serve as an indication of the location of ulcers. In a stomach filled with fluid or in an empty stomach, the location and pattern of the robot are identified by tracking the stacked white lines in the ultrasound images. The thickness of the stack decreases when a Type I robot transits into a Type III robot. In particular, the location where the white signal (the yellow dashed rectangles, Supplementary Figure 12) disappears can be used to locate the Type III robot in the air environment, allowing for tracking of the Type III robot as it moves in an in-vivo porcine stomach after gastric emptying.

Fig. R2 Characteristic ultrasound image of the robots and ulcers in different environments. **a**, Ulcers with different sizes on gastric tissue. **b**, Ultrasound images of the ulcers with different sizes in environments filled with water and air. The white dashed line and the blue dashed ellipses represent the gastric mucosa and the ulcer, respectively. The scale bar is 5 mm. **c**, Ultrasound images of the robot in environments filled with water and air. The blue dashed ellipses represent the gastric ulcer. The red dashed rectangles represent the robot. The yellow dashed rectangles represent white signal. The scale bar is 5 mm.

Comment #3: Pertaining to the *in vivo* study: Based on the photos, it appears some adhesives either delaminated or did not achieve optimal tissue contact. Could the authors shed light on how this might influence adhesion efficacy?

Response: Thank you for the comment. In our work, each layer of the robot is composed of a soft magnetic substrate and an embedded adhesive film, as shown in Fig. R3a. The delamination of the robot shown in the *in-vivo* results is attributed to the non-adhesive frame of the soft magnetic substrate, which in fact, does not affect the adhesion strength of the adhesive film on the tissue. In Fig. R3b, we prepare monomer samples with titled edges (Group 1), and then conduct reciprocal stretching experiments on these samples to evaluate their adhesion strength. The experimental result is shown in Fig. R3c. The retention time of Group 1 is 23.128 ± 1.138 h, which is only 11.34 % lower than that of the control group (26.098 ± 7.915 h). The result highlights that the delaminated edge of the frame will lead to minor influence of the adhesion performance of the adhesive film.

In Fig. R3b, we prepare monomer samples without frames (Group 2), and then conduct reciprocal stretching experiments on these samples to evaluate their adhesion strength. The experimental result is shown in Fig. R3c. The retention time of Group 2 is 9.993 ± 2.277 h, and it is 61.71 % lower compared to that of the control group. We believe the soft magnetic substrate with the frame can protect the adhesives after adhesion forms by providing an encapsulated layer to slow down water penetration.

Fig. R3 The influence of the frame on the adhesion efficacy. **a**, Fabrication of the layers of the robot. **b**, Schematics of monomers with different frames adhere with tissue. **c**, Retention time of different samples. The error bars are obtained from 3 trials in each condition.

Comment #4: *Could the authors elaborate on the limitations related to external control (distance, DoF) and the feasibility of translating this technology into clinical settings?*

Response: Thank you for the comments. In the current setup in this work, the limitation in distance of the effective magnetic maneuver is 20 mm, as shown in Fig. 4c and 4d. However, the effective operating distance can be further extended by increasing the strength of the magnetic field. This enhancement can be achieved with different methods, including increasing the current in the electromagnetic coil and using a stronger magnet. Regarding the degree of freedom, i.e., DoF, of the applied external control, we primarily consider the accessible regions of the robot actuated by external magnetic force and torque. With a sufficiently strong magnetic field, the robot can be actuated through magnetic field gradient forces towards any region inside the stomach. Additionally, by altering the direction of the magnetic field, the desired magnetic torque can be exerted to operate the robot effectively within the stomach, such as flipping motion and separate behaviour. Therefore, the degree of freedom of the external control system sufficient for the dexterous driving of the robot.

The feasibility of translating our technology into clinical scenario is supported by several factors. Both ex-vivo and in-vivo experiments have demonstrated that the proposed robot is capable of achieving multi-target adhesion in a three-dimensional non-structured environment filled with liquid and covered with a layer of mucus. Meanwhile, in-vitro cell toxicity experiments have confirmed the biocompatibility of the materials composing the robot. In addition, considering the specific needs required by a magnetic field actuation system for medical scenarios, the effective operation distance of the robot can be extended by increasing magnetic field strength and gradient. For systems integrated with permanent magnet, increasing the volume of the magnet can enhance both the magnetic field strength and gradient, while for electromagnetic coils, boosting the current intensity amplitude in the electromagnetic coils and implementing water-cooling equipment could also be a promising method. For the medical imaging in stomachs, we can apply this strategy under the guidance of X-ray imaging. Oral-taking barium meal can locate the ulcers (Debi, U. et al., *Trop. Gastroenterol.*, 2019), and the robot can be navigated to cover and adhere on them.

Reviewer 2:

The authors present a multi-layered soft robot where each layer is connected to another by means of magnetic forces. Additionally, each layer contains an adhesive film consisting of crosslinked Carbopol, Poloxamer, and HPMC that is able to form hydrogen bonds with the stomach lining. Movement and post-adhesion layer detachment of the robot is achieved by the application of magnetic fields. In order to achieve adhesion, magnetic forces exerted by an external magnet are used to press down the film on the stomach lining. The story of the manuscript is compelling and the graphic material including figures and videos are of a high visual and informative quality. However, I ask the authors to consider the following.

Response: Thank you for your overall detailed comments to our work. Your constructive comments are very helpful in improving the quality of our manuscript. We have addressed or made necessary discussions based on your comments point-by-point in the following part.

Comment #1: Regarding the clinical application. The adhesive layer is able to bond with gastric tissue and cover it, but does not appear to promote healing of the ulcer. Is there a specific clinical need for covering an ulcer or can the robot be extended with drugs to promote healing of the ulcer?

Response: Thank you for the comment. Covering an ulcer is an action that is needed in clinics (Bertleff, M. J. O. E. et al., *JSLs*, 13, 550-554, 2009; Xu, X. et al., *Sci. Transl. Med.*, 12, eaba8014, 2020), and a physical cover for an ulcer could protect it from the erosion caused by gastric juice and facilitate tissue generation (Chen, X. et al., *Adv. Funct. Mater.*, 32, 2202285, 2022; Peng, X. et al., *Sci. Adv.*, 7, eabe8739, 2021).

To validate the potential drug release capability of the robot, curcumin (Xu, L. et al., *Mol. Pharm.*, 20, 2105-2118, 2023), a drug to treat gastric ulcers, is loaded into the adhesive films. The in-vitro drug release of the adhesive films is measured by sample and separate method (D'Souza, *Adv. Pharm.*, 2014, 1-12, 2014) as shown in Fig. R4a. A side layer, i.e., comprises of a drug-loaded adhesive film and a soft magnetic substrate, is adhered to a gastric tissue with a hole, and they are placed on the top of the beaker consisting of 50 mL simulated gastric fluid (SGF). The release experiment is performed at 37 ± 0.5 °C, with a stirrer bar stirring at 50 rpm. At a predetermined time, 2 mL of solution is collected, and an equal volume of SGF is supplemented simultaneously. The collected sample solution is filtered through a filter paper and analyzed by a UV spectrophotometer at 420 nm. The cumulative release of curcumin is calculated as follows.

$$\text{Cumulative release percent (\%)} = \frac{C_n \times V_n + C_{n-1} \times V_{n-1} + \dots + C_1 \times V_1}{m_{cur}} \times 100\% \quad (1)$$

where C_n and C_{n-1} denote the concentration of curcumin released at times n and $n-1$, respectively.

V_n and V_{n-1} denote the collection volume at times n and $n-1$, respectively.

The result of drug release efficiency as shown in Fig. R4b. The cumulative release of curcumin gradually reaches approximately 80 % in 12 h. The results demonstrate that the robots are promising platforms for targeted delivery and sustained release of drugs for gastric ulcer treatment.

Fig. R4 In vitro drug release of the adhesive films. **a**, Schematics of sample and separate method. **b**, Cumulative release of curcumin loaded by the adhesive film. The error bars are obtained from 3 trials in each condition.

To clarify, we have revised the third paragraph in Section **Introduction**.

Magnetic soft robots offer an efficient and noninvasive approach for delivering bioadhesive platforms to gastric ulcers, which can mitigate erosion and enhance the healing effect by covering gastric ulcers^{42,43}.

Comment #2: Regarding magnetic actuation. Based on the design of the robot, there appears to be a missing characterization (can also be based on simulation) of the layer-layer interaction forces and torques during separation. In addition, this should be connected to the required magnetic field to achieve separation and placed in context with magnetic fields typically achievable by state-of-the-art medical-purpose magnetic field generating systems (e.g., DOI 10.1002/adfm.202005137).

Response: Thank you for the comment. Herein, simulation of layer-layer interaction forces and torques during the separation process is conducted and comprehensively discussed. The separation process is realized by changing the applied magnetic field direction, which is defined as the angle between the magnetic field and the z-axis, and keeping the magnetic field strength unchanged. The simulation results and related analysis of the layer-layer interaction forces and torques exerted for its separation are shown and discussed (Fig. R5a). The simulated stress distribution shown in the color maps of Fig. R5a indicates that, increasing the magnetic field direction can result in a large deformation of the robot, contributing to the flipping and separation behaviour. The relationship between the layer-layer interaction forces, toques and the magnetic field direction is presented in Fig. R5b, R5c, and R5d. With the change in the applied magnetic field direction, the horizontal force F_x exerted between layers gradually increases, while the vertical force F_z gradually decreases. The magnetic torque τ_z exerted about the z-axis direction also decreases.

Considering the specific needs required by a magnetic field actuation system for medical scenarios, the space for effective operation of the multi-layer robot shall reach $200 \times 200 \times 60 \text{ mm}^3$. The space guarantees the entire coverage of the stomach region of a human. Simulation results presented in Fig. 5a, b reveal that when the magnetic field strength reaches 50 mT, the generated magnetic torque can effectively disrupt the layer-layer interaction, enabling the layer-layer separation. Meanwhile, the demanded magnetic field strength is reachable by a magnetic actuation system (Zhao, A.-J. et al., *Gastrointest. Endosc.*, 88, 466-474, 2018; Liao, Z. et al., *Clin. Gastroenterol. Hepatol.*, 14, 1266-1273, 2016; Xu, Y. et al., *IEEE Trans. Autom. Sci. Eng.*, 18, 1640-1652, 2021). Therefore, our approach exhibits significant potential for clinical translation.

Fig. R5 Simulation results of the layer-layer interaction forces and torques during separation. **a**, Simulation of the separation process of a soft magnetic film actuated by a magnetic field with changed field direction (from 0° to 80°) and constant magnetic field strength. **b**, Simulation results of the layer-layer interaction force F_x exerted on the separated film with the change of the applied magnetic field direction. **c**, Simulation results of the layer-layer interaction force F_z exerted on the separated film with the change of the applied magnetic field direction. **d**, Simulation results of layer-layer interaction torque τ_z exerted on the separated film with the change of the applied magnetic field direction.

In order to give more details to characterize the layer-layer interaction forces and torques during the separation process, we have added the results presented in Fig. R5b, R5c and R5d into the supplementary information, as Supplementary Fig. 8c. We have revised the first paragraph in Section **On-demand separation by flipping motion** as follows: “The relationship between the layer-layer interaction forces, torque, and the magnetic field direction during separation is present in Supplementary Fig. 8c, showing that horizontal force increases with the field direction, while the vertical force and magnetic torque gradually decrease.”.

A new reference is cited: “6. Ebrahimi, N. et al. Magnetic Actuation Methods in Bio/Soft Robotics. *Adv. Funct. Mater.* **31**, 2005137 (2021).”.

Comment #3: Continuing on my previous comment, based on Fig. 4c the magnetic force of 6N exerted on the robot by the external permanent magnet seems high when comparing to other works reporting forces on magnetic polymers by magnets (e.g., DOI 10.1089/soro.2022.0031). I am missing some validation regarding the actual forces exerted by the magnet on the robot, as well as a quantification of the necessary field gradients. In reality, I assume a magnet-robot distance of 15mm is unachievable at all times around the stomach, and any increase in distance has a significant effect on the achievable magnetic forces on the robot.

Response: Thank you for your helpful and detailed comments. As you mentioned, the force magnitude reported in the last version is unachievable. After revision, we recalculate the maximum magnetic force exerted on the robot (Type I robot) by the external permanent magnet, and the value is 0.364 N with a vertical distance of 15 mm (Fig. R6a). The maximum magnetic force exerted on the Type II and Type III robots are 0.316 N and 0.209 N with a vertical distance of 15 mm (Fig. R6b and c). The magnetic force exerted on the robot gradually decreases as the vertical distance increases. When the vertical distance is 30 mm, the superposed external force is zero so the robot remains static.

Fig. R6 Simulation results and analysis of the forces exerted on the robot for its translational motion on gastric tissue. **a-c**, Relationships between the simulated superposed external force exerted on (a) a Type I robot, (b) a Type II robot and (c) a Type III robot performing translational motion and the horizontal distance L . Different vertical distances d are applied.

The solid lines represent the magnetic force F_m^x , and the dashed lines represent the friction force F_f .

To validate and quantify the actual forces exerted by the magnet on the robot, as well as the necessary field gradients, we have measured the magnetic field strength of a cylindrical permanent magnet (a cross-sectional diameter of 30 mm and a height of 30 mm) by using a Hall sensor (Fig. R7a). The magnetic field strength B_x initially increases and then decreases with the increase of the horizontal distance. The

magnetic field gradient $\frac{\partial B_x}{\partial z}$ is obtained by derivation of the magnetic field strength B_x as shown in

Fig. R7b. The horizontal magnetic force F_x^m exerted on the robot (Type I robot) by the external permanent

magnet can be expressed as Eq. 2. The results demonstrate that the horizontal magnetic force F_x^m initially increases and then decreases with the increase of the horizontal distance as shown in Fig. R7c.

$$F_m = \begin{bmatrix} F_x^m \\ F_y^m \\ F_z^m \end{bmatrix} = \begin{bmatrix} \frac{\partial B_x}{\partial x} & \frac{\partial B_x}{\partial y} & \frac{\partial B_x}{\partial z} \\ \frac{\partial B_y}{\partial x} & \frac{\partial B_y}{\partial y} & \frac{\partial B_y}{\partial z} \\ \frac{\partial B_x}{\partial z} & \frac{\partial B_y}{\partial z} & -\left(\frac{\partial B_x}{\partial x} + \frac{\partial B_y}{\partial y}\right) \end{bmatrix} \begin{bmatrix} m_x \\ m_y \\ m_z \end{bmatrix} = \begin{bmatrix} \frac{\partial B_x}{\partial x} & 0 & \frac{\partial B_x}{\partial z} \\ 0 & 0 & 0 \\ \frac{\partial B_x}{\partial z} & 0 & -\frac{\partial B_x}{\partial x} \end{bmatrix} \begin{bmatrix} 0 \\ 0 \\ m_z \end{bmatrix} = \begin{bmatrix} \frac{\partial B_x}{\partial z} \cdot m_z \\ 0 \\ -\frac{\partial B_x}{\partial x} \cdot m_z \end{bmatrix} \quad (2)$$

The effective operation distance of the magnetic actuation system is important. An increase in magnetic field strength and gradient can extend the effective distance to operate the robot using the system. The magnetic actuation system with a relatively small cylindrical magnet (a cross-sectional diameter of 30 mm and a height of 30 mm) is used in this study. The maximum distance of this system for effective operating the robot is 20 mm. At operating distance of 20 mm, the magnetic field strength is 22.86 mT and the magnetic field gradient is $-3.452 \text{ kg}/(\text{m} \cdot \text{s}^2 \cdot \text{A})$. To further elongate the effective operational distance, we enhance the setup by using a larger magnet (a cross-sectional diameter of 50 mm and a height of 100 mm). Experimental results demonstrate that the maximum operating distance of the robot is increased to 55 mm, the magnetic field strength is 22.16 mT and the magnetic field gradient is $-0.8127 \text{ kg}/(\text{m} \cdot \text{s}^2 \cdot \text{A})$ (Fig. R7d). Therefore, the upgrading of the magnetic actuation system on field strength and field gradient is critical for further extending its effective distance to maneuver the robot. For systems with permanent magnet,

increasing the volume of the magnet can enhance both the magnetic field strength and gradient, while for electromagnetic coils, boosting the current amplitude in the coil and implementing water-cooling equipment could also be a promising method.

Fig. R7 Characterization of actual magnetic strength, gradient, and force exerted by the magnet on the robot. **a**, Actual magnetic strength B_x exerted by the magnet on the robot. **b**, Actual magnetic gradient $\frac{\partial B_x}{\partial z}$ exerted by the magnet on the robot. **c**, Actual magnetic force F_x^m exerted by the magnet on the robot. **d**, Translational motion of the robot on the gastric tissue at a vertical distance of 55 mm.

We have replaced Fig. 4c with Fig. R6a in the manuscript, and Supplementary Fig. 4 with Fig. R6b and c. We have revised the first paragraph in Section **Magnetic actuation of the robot**. We have revised the Section **Discussion** by adding “Considering the specific needs required by a magnetic field actuation system for medical scenarios, the effective operation distance of the robot can be extended by increasing magnetic field strength and gradient. For systems integrated with permanent magnet, increasing the volume of the magnet can enhance both the magnetic field strength and gradient, while for electromagnetic coils, boosting the current amplitude in the electromagnetic coil and implementing water-cooling equipment could also be a promising method.”.

In Supplementary Information, we have revised the Supplementary Table S1.
Supplementary Table S1

The effective range of the horizontal distance L , at which the magnetic multi-layer soft robot can be actuated on the gastric tissue. Different vertical distances d are applied.

	Horizontal distance L /mm			
	$d = 15$ mm	$d = 20$ mm	$d = 25$ mm	$d = 30$ mm
Type I Robot	$3.5 < L < 36$	$5.5 < L < 33.5$	$10 < L < 29$	/
Type II Robot	$4.5 < L < 33$	$7.5 < L < 30.5$	$14.5 < L < 23$	/
Type III Robot	$5 < L < 29$	$14 < L < 19.5$	/	/

A new reference is cited: “11. Richter, M. et al. Magnetic Soft Helical Manipulators with Local Dipole Interactions for Flexibility and Forces. *Soft Robot.* **10**, 647–659 (2023).”.

Comment #4: *The experimental procedure can be elaborated more. Was the ultrasound transducer operated manually as well as the magnet, simultaneously by one person?*

Response: Thank you for the comments. In the experiment, both the transducer and magnet are manually operated. Two operators collaborate to achieve multi-target adhesion of the robot. Operator 1 operates the transducer to maintain the continuous contact with the outer gastric wall, in order to locate the robot in the stomach using real-time ultrasound imaging. Once the robot is located, Operator 2 positions the magnet close to the transducer, which can attract the robot to contact the gastric wall. Operators 1 and 2 move the transducer and magnet synchronously toward target I. Upon target I appearing in the ultrasound image, Operator 1 fixes the transducer, while Operator 2 adjusts the position of the robot to cover the ulcer precisely by moving the magnet. Subsequently, the adhesion between the adhesive film and the tissue is formed under magnetic attraction. Operator 2 then reverses the magnet to separate Type II robot from the adhered side layer. This systematic approach is also applied to cover targets II and III.

It is noted that while our experiments involved manual operation, the same methodology can translate into fully automated setup. By employing two robot arms to combine with the ultrasound transducer and the magnet respectively, automated guidance operations can be achieved in the same way.

To supplement details of the operation, we have added a Section in **Methods** about experimental procedure of ex-vivo and in-vivo tests as follow.

Experimental procedure of ex-vivo and in-vivo tests

Experimental procedures of ex-vivo and in-vivo experiments can be listed as follows. Two operators collaborate to achieve multi-target adhesion of the robot. Operator 1 operates the ultrasound transducer to locate the robot. Once the robot is located, Operator 2 manipulates the robot using a cylindrical magnet. Operators 1 and 2 move the transducer and magnet synchronously to actuate the robot toward and then cover target I. Subsequently, once the adhesion between the adhesive film and the tissue is formed under magnetic attraction. Operator 2 then reverses the magnet to separate Type II robot from the adhered side layer. This systematic approach is also applied to cover targets II and III.

Comment #5: *The ex vivo demonstrations are compelling. However, what do the subsequent in vivo experiments aim to show in addition to the ex vivo experiments, considering the animal is euthanized post-experiment and the stomach dissected to again inspect ex vivo? I acknowledge that the authors mention that the in-vivo gastric environment is more complex with secretion of mucus on the gastric mucosa. An ex vivo stomach is structurally similar to an in vivo stomach, hence an elaboration of the added complexity is warranted. Similarly, is the secretion of mucus on the gastric mucosa a concern, in the sense that it may increase friction?*

Response: Thank you for the comment. In-vivo stomachs add challenges and complexities compared with ex-vivo ones. While the in-vivo stomach and the ex-vivo stomach are structurally similar, the internal environment of the living porcine stomach brings significant complexities, such as the dynamic peristalsis and the consistent secretion of gastric mucus. The peristaltic motion of the living stomach poses challenges in monitoring and precisely locating the robot within the gastric environment. Additionally, the continuous secretion of gastric mucus makes effective contact difficult between the adhesive film and the tissue, which

could prevent the formation of stable adhesion. Furthermore, the in-vivo stomach has more folds on the surface. Therefore, in-vivo experiments are necessary. It is noteworthy that the presence of mucus does not increase the friction encountered by the robot during the movement along the mucosa, which is a promising advantage of the robot in physiological conditions.

In order to clarify this point, we have revised the Section **Multi-target adhesion in an in-vivo porcine stomach** by adding “**In-vivo stomachs add challenges and complexities compared with ex-vivo ones. While the in-vivo stomach and the ex-vivo stomach are structurally similar, the internal environment of the living porcine stomach brings significant complexities, such as the dynamic peristalsis and the consistent secretion of gastric mucus. The peristaltic motion of the living stomach poses challenges in monitoring and precisely locating the robot within the gastric environment. Additionally, the continuous secretion of gastric mucus makes effective contact difficult between the adhesive film and the tissue, which could prevent the formation of stable adhesion. Furthermore, the in-vivo stomach has more folds on the surface. Therefore, in-vivo experiments are necessary.**”.

Comment #6: *Based on my understanding of the visual material, it appears the experiments were performed with the gastric mucosa below the robot in the direction of gravity. How will the robot work if an ulcer is located on the upper gastric wall. If the answer is magnetic attraction, what is the weight of the robot and which field gradients are necessary to achieve adherence against the direction of gravity (based on simple calculations)?*

Response: Yes, our experiment are performed with the gastric mucosa below the robot in the direction of gravity. When the ulcer is located on the upper gastric wall, the robot can be attracted by magnetic forces and then guided to the ulcer. To evaluate the feasibility of this method, we calculate the magnetic force exerted on the robot by a magnet placed at different vertical distances using Eq. 3.

$$F_m = \begin{bmatrix} F_x^m \\ F_y^m \\ F_z^m \end{bmatrix} = \begin{bmatrix} \frac{\partial B_x}{\partial x} & \frac{\partial B_x}{\partial y} & \frac{\partial B_x}{\partial z} \\ \frac{\partial B_y}{\partial x} & \frac{\partial B_y}{\partial y} & \frac{\partial B_y}{\partial z} \\ \frac{\partial B_z}{\partial x} & \frac{\partial B_z}{\partial y} & \frac{\partial B_z}{\partial z} \end{bmatrix} \begin{bmatrix} m_x \\ m_y \\ m_z \end{bmatrix} = \begin{bmatrix} 0 & 0 & 0 \\ 0 & 0 & 0 \\ 0 & 0 & \frac{\partial B_z}{\partial z} \end{bmatrix} \begin{bmatrix} 0 \\ 0 \\ m_z \end{bmatrix} = \begin{bmatrix} 0 \\ 0 \\ \frac{\partial B_z}{\partial z} \cdot m_z \end{bmatrix} \quad (3)$$

We measure the magnetic field strength of a cylindrical permanent magnet (a cross-sectional diameter of 50 mm and a height of 50 mm) by using a Hall sensor (Fig. R8a). The magnetic field strength B_z increases with the decrease of the vertical distance. The magnetic field gradient $\frac{\partial B_z}{\partial z}$ is obtained by derivation of the magnetic field strength B_z as shown in Fig. R8a. Based on the measured magnetic field strength B_z and calculated magnetic field gradient $\frac{\partial B_z}{\partial z}$, the magnetic force F_z^m exerted on the robot during the descent of the magnet is calculated, as shown in Fig. R8b. The magnetic force F_z^m increases with the decrease of the vertical distance. When the vertical distance is 61 mm, the magnetic force surpasses gravity. The magnetic field strength at this condition is 21.005 mT, and the magnetic field gradient is $-0.332 \text{ kg}/(\text{m}\cdot\text{s}^2\cdot\text{A})$.

The schematics of the lifting motion of the robot is shown in Fig. R8c. As shown in Fig. R8d, experimental results demonstrate that when the vertical distance is 60 mm, the robot can overcome the gravity and contact with the upper gastric wall under magnetic attraction. After the magnet is removed, the robot adheres to the gastric tissue. In this experiment, the mass of the robot is measured to be 0.601 ± 0.054 g.

Fig. R8 Lifting motion of the robot. **a**, Magnetic strength B_z and gradient $\frac{\partial B_z}{\partial z}$ of a cylindrical magnet with a diameter is 50 mm and a height is 50 mm at different vertical distances. **b**, Relationship between the superposed external force exerted on the robot (Type I robot) performing lifting motion at different vertical distances. The blue solid lines represent the magnetic force F_z^m , and the red dashed lines represent the gravity G . **c**, Schematics of lifting motion of the robot. **d**, Experimental results of the lifting motion of the robot actuated by a magnet. The orange arrows represent the motion direction of the magnet.

Comment #7: Reference 18 contains n/a

Response: We have revised the Reference 18 in the manuscript.

20. Wang, C., Mzyk, A., Schirhagl, R., Misra, S. & Venkiteswaran, V. K. Biocompatible Film-Coating of Magnetic Soft Robots for Mucoadhesive Locomotion. *Adv. Mater. Technol.* **8**, 2201813 (2023).

Response to Reviewer 3:

This paper presents a magnetic soft robot consisting of multiple layers with distinct in-plane structures and functions to achieve on-demand adhesion to different lesions. In addition, it demonstrates two locomotion modes and the separation with one side layer adhered to tissues. However, the following issues should be considered and addressed in the revised manuscript.

Response: We thank the reviewer for the constructive comments on our work. We have carefully addressed all the questions raised by the reviewer point-by-point and revised our manuscript accordingly.

Comment #1: *If the soft robot is placed on the stomach surface for more than a certain period of time without magnetic force applied to the robot, does not the adhesion between each layer and the stomach surface occur? If so, is there any possibility that this will lose the adhesion of the layer?*

Response: Thank you for the comment. If the soft robot is placed on the tissue surface without applying magnetic attraction, it will not form a strong adhesion with the tissue, and the subsequent adhesion and separation are still feasible.

As depicted in Fig. R9, the robot (Type I robot) is first placed on the tissue surface without applying magnetic attraction. After 5 minutes, the robot can be easily actuated from its initial position to the target position by applying magnetic field gradient. Subsequently, by applying magnetic pressing forces to the robot for another 5 minutes, the robot cannot be translated from the target to another position anymore, indicating that the adhesion is formed between the side layer of the robot and the tissue. The Type II robot can then successfully separate from the side layer by reversing the direction of the magnetic field. Experimental results demonstrate that placing the robot on the gastric tissue for 5 minutes without magnetic force does not affect the adhesion performance of the adhesive film.

Fig. R9 The adhesion situation between the adhesive film and the tissue with and without magnetic pressing forces. The orange arrow represents the translational motion direction of the robot.

Comment #2: *Detailed information such as dimensions, weight, etc. of the proposed robot should be provided. In addition, the robot is expected to be light because it consists of three thin layers. So, it is expected that the flow of gastric fluid affects the robot's movement.*

Response: Thank you for the comment. The detailed information about the robots we utilized in this study

is presented in Table R1. Although the weight of the robot is light (mass of the Type I robot is 0.612 ± 0.054 g), the flow of gastric fluid could still be a minor factor in affecting the robot's movement. We expect the flow of gastric fluid is mainly induced by the gastric peristalsis. Since the frequency of gastric peristalsis is 0.05 Hz (Marciani, L. et al., *Am. J. Physiol.-Gastrointest. Liver Physiol.*, 280, G844-G849, 2001; Schulze, K., *Neurogastroenterol. Motil.*, 18, 172-183, 2006) and the operational frequency of the robot significantly surpasses this peristalsis frequency. Meanwhile, after gastric emptying, the liquid left inside the stomach is minimal. Therefore, the movement of the robot inside the stomach could remain unaffected. Additionally, the in-vivo experiments presented in this work have also demonstrated the capability of the robot in performing targeted locomotion inside a porcine stomach.

In order to giving more details about the robot, we have added Table R1 to supplementary information, as Supplementary Table S2. We have revised the **Preparation of porcine stomach for ex-vivo tests** in Section **Methods** by adding “For ex-vivo tests, the dimensions of the robot (Type I robot) are $18 \times 18 \times 1.5$ mm³ and the weight is 0.612 ± 0.054 g.”. We have revised the **In-vivo tests in a pig** in Section **Methods** by adding “For in-vivo tests, the dimensions of the robot (Type I robot) are $18 \times 18 \times 1.5$ mm³ and the weight is 0.612 ± 0.054 g.”.

Table R1

Detailed information of the proposed robot.

	Type I robot	Type II robot	Type III robot
Weight (g)	0.612 ± 0.054	0.412 ± 0.025	0.221 ± 0.007
Length \times Width \times Thickness (mm ³) of the robot	$18 \times 18 \times 1.5$	$18 \times 18 \times 1.1$	$18 \times 18 \times 0.7$
Length \times Width \times Thickness (mm ³) of adhesive film	$15 \times 15 \times 0.08$	$15 \times 15 \times 0.08$	$15 \times 15 \times 0.08$

Comment #3: *The proposed robot should be loaded with drugs. It should also show how efficiently the drug can be loaded, how effectively the drug is released, and the performance of the treatment by the released drug. You can also load a model drug rather than a real drug. In addition, since loading drugs can affect the mobility or adhesion of the proposed robot, the related experiments should be included or discussed.*

Response: Thank you for the comment. Since the adhesive films are polymer networks, drugs can be loaded. To evaluate the drug loading and release capability, we have prepared adhesive films loaded with curcumin (Xu, L. et al., *Mol. Pharm.*, 20, 2105-2118, 2023), i.e., a drug to treat gastric ulcers, with different concentrations, ranging from 5 % to 25 %. The adhesion strength of adhesive films loaded with the drugs is evaluated through Lap shear tests (Wu, J. et al., *Sci. Transl. Med.*, 14, eabh2857, 2022; Yuk, H. et al., *Nature*, 575, 169-174, 2019). The tested shear strength, i.e., the strength of the adhesion between the adhesive film and the tissue, is presented in Fig. R11a. The figure also indicates the relationship between the adhesion strength and the drug-loaded ratio, i.e., the ratio of the drug mass and the mass of the whole film. The experimental results demonstrate that the adhesion strength decreases with the increase of the curcumin content. When the drug-loaded ratio is 5 % (w/w), the shear strength of the adhesive film decreases by 26.87 % compared to that without loading drug. When the drug-loaded ratio of the adhesive film increases from 5 % to 20 %, shear strength only decreases by 6.14 %. Moreover, the robot fabricated with a 25 % curcumin load can still complete multi-target adhesion to ex-vivo porcine gastric tissue (Fig. R11b). A Type I robot is delivered to Target I actuated by an external magnetic field ($t = 6$ s), and the adhesion between the side layer of the robot and the gastric tissue is formed by applying a vertical magnetic attraction force for approximately 2 mins. Subsequently, a Type II robot separates from the adhered side layer by changing the direction of the magnetic field ($t = 155$ s) and is navigated to Target II to adhere to it

($t = 165$ s). Finally, a Type III robot separates from the second adhered side layer ($t = 280$ s) and moves to Target III for the third adhesion ($t = 304$ s).

Fig. R11 Characterization of adhesion performance and drug release of the drug-loaded adhesive films. **a**, Relationship between shear strength of the adhesive films and the drug-loaded ratio. The error bars are obtained from 3 trials in each condition. **b**, Experimental results of the multi-target adhesion using the robot loaded 25 % curcumin on ex-vivo gastric tissue. The scale bar is 15 mm. **c**, Schematics of sample and separate method. **d**, Cumulative release of curcumin loaded by the adhesive film. The error bars are obtained from 3 trials in each condition.

The in-vitro drug release of the adhesive films is measured by sample and separate method (*D'Souza, Adv. Pharm., 2014, 1-12, 2014*) as shown in Fig. R11c. A side layer, i.e., comprises of a drug-loaded adhesive film and a soft magnetic substrate, is adhered to a gastric tissue with a hole, and they are placed on the top of the beaker consisting of 50 mL simulated gastric fluid (SGF). The release experiment is performed at 37 ± 0.5 °C, with a stirrer bar stirring at 50 rpm. At a predetermined time, 2 mL of solution is collected, and an equal volume of SGF is supplemented simultaneously. The collected sample solution is filtered through a filter paper and analyzed by a UV spectrophotometer at 420 nm. The cumulative release of curcumin is calculated as follows.

$$\text{Cumulative release percent (\%)} = \frac{C_n \times V_n + C_{n-1} \times V_{n-1} + \dots + C_1 \times V_1}{m_{cur}} \times 100\% \quad (4)$$

where C_n and C_{n-1} denote the concentration of curcumin released at times n and $n-1$, respectively.

V_n and V_{n-1} denote the collection volume at times n and $n-1$, respectively.

The experimental results of curcumin release efficiency as shown in Fig. R11d. The cumulative release of curcumin gradually reaches approximately to 80 % in 12 h. The results demonstrate that the robots are promising platforms for targeted delivery and sustained release of drugs for gastric ulcer treatment.

The proposed robot has the potential in effectively treat ulcers, and the reasons are two-fold. First, the robot has a strong capability in targeted delivery and adhesion. Gastric juice could be a major factor that hinders the healing of gastric ulcers (*Shan, J. et al., Mater. Sci. Eng. C, 103, 109870, 2019; Zhao, L.-M. et al., Regen. Biomater., 8, rbaa056, 2021*). The adhesion provided by the multi-layer robots can effectively protect the ulcers from further erosion caused by the gastric juice, which could benefit the healing process (*Chen, X. et al., Adv. Funct. Mater., 32, 2202285, 2022; Peng, X. et al., Sci. Adv., 7, eabe8739, 2021*). Second, curcumin is a widely-used drug to treat gastric ulcers (*Xu, L. et al., Mol. Pharm., 20, 2105–2118, 2023; Omidian, H. et al., Gels 9, 596, 2023*), and the robot loaded the drug with a high capacity can precisely adhere to the targeted ulcers. The sustained releasing capability of the robot is validated in this work (Fig. R11d), providing the possibility of a faster healing rate. In fact, each layer of the robot comprises a soft magnetic substrate and an embedded drug-loaded adhesive film. The soft magnetic substrate can also serve as structure encapsulating and protecting the drug-loaded adhesive film from undesired swelling due to the existence of the gastric juice, which can further prolong the adhesion and drug release time period at the site of the ulcers.

To evaluate the influence of drug-loaded behaviour on the mobility of the robot, a robot with 5 % curcumin is fabricated for the mobility test. The robot realizes translational motion overcoming the friction force between the robot and the tissues at a vertical distance of 20 mm actuated by magnetic force as shown in Fig. R12a. The tumbling motion of the Type I robot under a rotating magnetic field with a strength of 20 mT and a frequency of 0.2 Hz is shown in Fig. R12b. Experimental results demonstrate that the robot with more layers can reach a higher step-out frequency at the same field strength ($B = 20$ mT). Moreover, the translational speed of the robot performing tumbling motion increases with the applied field frequency initially, and then decreases after the field frequency reaches the step-out frequency (Fig. R12c). The experimental results demonstrate that the step-out frequency of the robot that loaded the drug is consistent with the robot without the drug. Hence, loading drugs does not affect the mobility of the proposed robot.

In order to give more details about the drug loaded robot, we have added the results presented in Fig. R11 into the supplementary information, as Supplementary Fig. 15. We have added the results presented in Fig. R12 into the supplementary information, as Supplementary Fig. 16. We have revised the last paragraph in Section **Discussion** by adding “In addition, loading drugs into the adhesive film of the robot will help further transform the technology into clinical applications that maintain its own movement and adhesion performance (Supplementary Fig. 15 and Supplementary Fig. 16).”.

Fig. R12 Locomotion of the drug-loaded robots. **a**, Translational motion of the robot on the gastric tissue at a vertical distance of 20 mm. The white arrows represent the translational motion direction of the magnet. **b**, Tumbling motion of the robot on gastric tissue. **c**, Relationship between the speed of the robot performing tumbling motion and the frequency of the magnetic field. The error bars are obtained from 3 trials in each condition.

Comment #4: *It appears that a permanent magnet was used to move the robot. Is the magnet controlled manually? Or do you use a macro robot arm for the locomotion of the soft robot? Also, when the layer is separated by magnetic force, how is the magnetic force controlled at this time? Finally, the information on how the robot is controlled and the actuation system should be included in the manuscript.*

Response: Yes, in both ex-vivo and in-vivo experiments, manual operation is utilized for controlling the permanent magnets. In the experiments, both the transducer and magnet are manually operated. Two operators collaborate to achieve multi-target adhesion of the robot. Operator 1 operates the transducer to maintain the continuous contact with the outer gastric wall, in order to locate the robot in the stomach using real-time ultrasound imaging. Once the robot is located, Operator 2 positions the magnet close to the transducer, which can attract the robot to contact the gastric wall. Operators 1 and 2 move the transducer and magnet synchronously toward target I. Upon target I appearing in the ultrasound image, Operator 1 fixes the transducer, while Operator 2 adjusts the position of the robot to cover the ulcer precisely by moving the magnet. Subsequently, the adhesion between the adhesive film and the tissue is formed under magnetic attraction. Operator 2 then reverses the magnet to separate Type II robot from the adhered side layer. This systematic approach is also applied to cover targets II and III.

It is noted that while our experiments involved manual operation, the same methodology can translate into fully automated setup. By employing two robot arms to combine with the ultrasound transducer and the

magnet respectively, automated guidance operations can be achieved in the same way.

A macro robot arm with a permanent magnet as the end effector can also actuate the multi-layer robot for navigated locomotion (Fig. R13a). During the actuation, the vertical distance between the robot and the magnet maintains 15 mm. The robot can also achieve climbing motion on a 45° slope on the gastric tissue through magnetic guidance (Fig. R13b). The targeted locomotion and separation of a Type I robot are shown in Fig. R13c, actuated by a permanent magnet linked on a robot arm. From 0 s to 150 s, the robot is navigated to cover the target through translational motion. The adhesive film of the side layer forms adhesion with gastric tissue under the magnetic force ($t = 800$ s). The Type II robot separates from the adhered side layer through reversing the magnetic field direction by rotating the last joint of the robot arm ($t = 801$ s). Therefore, even though this study manually operate the magnet for the actuation of the robot, a programmed motion of a robot arm integrated with the magnet can well serve to perform the same tasks.

Fig. R13 Locomotion of the robot operated by a robot arm with a magnet. **a**, Translational motion of the robot. The orange arrows represent the translational motion direction of the magnet. **b**, Climbing motion of the robot on a 45° slope. **c**, Targeted navigation and separation of the robot from the adhered side layer.

For the separation process, magnetic torque is leveraged to flip the robot over. The magnetic torque can be expressed as $\tau = m \times B$. The magnitude and direction of the magnetic torque can be controlled by changing the strength and direction of the magnetic field. By applying a magnetic actuation setup with a permanent magnet, the magnitude of the force and torque exerted on the robot can be tuned with the robot-magnet distance, while the direction of the force and torque can be changed with the rotational direction of the magnet. Moreover, the simulation results of layer-layer separation behaviour demonstrate that, the robot separates from the adhered side layer when the field strength is 50 mT (Fig. 5b), which is achievable by the setup. As aforementioned, we have demonstrated a robot can be actuated by a robot arm loaded with a

cylindrical magnet (a cross-section diameter of 50 mm and a height of 50 mm) in Fig. R13.

We have added the results presented in Fig. R13 into the supplementary information, as Supplementary Fig. 17. We have revised the last paragraph at Section **Discussion** by adding “It is noted that, a robot arm with a permanent magnet serving as its end effector can also realize the on-demand actuation of the robots (Supplementary Fig.17).”.

To evaluate the details during the experimental operation, we have added a Section in **Methods** about the experimental procedure of the ex-vivo and in-vivo tests.

Experimental procedure of ex-vivo and in-vivo tests

Experimental procedures of ex-vivo and in-vivo experiments can be listed as follows. Two operators collaborate to achieve multi-target adhesion of the robot. Operator 1 operates the ultrasound transducer to locate the robot. Once the robot is located, Operator 2 manipulates the robot using a cylindrical magnet. Operators 1 and 2 move the transducer and magnet synchronously to actuate the robot toward and then cover target I. Subsequently, once the adhesion between the adhesive film and the tissue is formed under magnetic attraction. Operator 2 then reverses the magnet to separate Type II robot from the adhered side layer. This systematic approach is also applied to cover targets II and III.

REVIEWERS' COMMENTS

Reviewer #1 (Remarks to the Author):

I thank the authors for the revised manuscript and my concerns have been addressed in full. I recommend publication of this work.

Dear Editor and Reviewers:

We thank the support from the reviewers and the editor. The response to the reviewer is listed.

Reviewer 1:

I thank the authors for the revised manuscript and my concerns have been addressed in full. I recommend publication of this work.

Response: We appreciate the reviewer for the time and support on our manuscript.